# Multi-instrument observations of polar cap patches and traveling ionospheric disturbances generated by solar wind Alfvén waves coupling to the dayside magnetosphere

Paul Prikryl[1], Robert G. Gillies[2], David R. Themens[1,3], James M. Weygand[4], Evan G. Thomas[5]
Shibaji Chakraborty[6]

1Physics Department, University of New Brunswick, Fredericton, NB, E3B 5A3, Canada

2Department of Physics and Astronomy, University of Calgary, Calgary, AB, Canada

3School of Engineering, University of Birmingham, Birmingham, UK

4Earth, Planetary, and Space Sciences, University of California, Los Angeles, CA, USA

5Thayer School of Engineering, Dartmouth College, Hanover, NH, USA

6Bradley Department of Electrical and Computer Engineering, Virginia Tech, Blacksburg, VA, USA

*Correspondence to*: Paul Prikryl (paul.prikryl@unb.ca)

**Abstract.** During minor to moderate geomagnetic storms, caused by corotating interaction regions at the leading edge of high-speed streams, solar wind Alfvén waves modulated the magnetic reconnection at the dayside magnetopause. The Resolute Bay Incoherent Scatter Radars (RISR-C and RISR-N), measuring plasma parameters in the cusp and polar cap, observed ionospheric signatures of flux transfer events that resulted in the formation of polar cap patches. The patches were observed as they moved over the RISR, and the Canadian High-Arctic Ionospheric Network (CHAIN) ionosondes and GPS receivers. The coupling process modulated the ionospheric convection and the intensity of ionospheric currents, including the auroral electrojets. The horizontal equivalent ionospheric currents are estimated from ground-based magnetometer data using an inversion technique. Pulses of ionospheric currents that are a source of Joule heating in the lower thermosphere launched atmospheric gravity waves, causing traveling ionospheric disturbances (TIDs) that propagated equatorward. TIDs were observed in the SuperDual Auroral Radar Network (SuperDARN) HF radar ground scatter and the detrended total electron content measured by globally distributed Global Navigation Satellite System (GNSS) receivers.

## 1 Introduction

Solar wind coupling to the dayside magnetosphere (Dungey, 1961, 1995) generates variable electric fields that map to the cusp ionosphere, driving the ionospheric convection and currents. The magnetic reconnection on the dayside magnetopause leads to open magnetic flux carried over the polar cap to the magnetotail (Dungey, 1961; Tsurutani and Meng, 1972; Russell and Elphic, 1978, 1979; Provan et al., 1998). The transient nature of magnetic reconnection at the dayside magnetopause is exemplified by flux transfer events (FTEs) (Russel and Elphic, 1978) and their ionospheric signatures have been extensively

studied (Van Eyken et al., 1984; Goertz et al., 1985; Southwood, 1987; Pinnock et al., 1995; Rodger et al., 1997; Provan et al. 1998, and references therein). Provan et al. (1998) observed a series of quasiperiodic pulsed azimuthal flow transients poleward of the convection reversal boundary (CRB), which they identified as the ionospheric signatures of FTEs.

The FTE signatures in the cusp ionospheric flows were proposed to be studied with the incoherent scatter radar (ISR) (Cowley et al., 1990), and were later observed by the coherent scatter radars of the SuperDual Auroral Radar Network (SuperDARN) (Pinnock et al., 1993, Provan et al., 1998), as well as the ISR in Svalbard (Oksavik et al., 2006). The Resolute Bay Incoherent Scatter Radars, measuring ionospheric plasma parameters in the cusp and polar cap are well suited to observe the FTE signatures (Gillies et al., 2016; 2018).

Cowley and Lockwood (1992) proposed that time-dependent magnetic reconnection and the resulting convection produce polar cap patches from dayside enhanced ionospheric density of a tongue of ionization (TOI) that is drawn through the cusp into the polar cap. Ionospheric flow channels, primarily in the F-region ionosphere, produce depletions in the ionospheric plasma, segmenting a TOI into patches (Pinnock et al., 1993, Rodgers et al., 1994; Valladares et al., 1994; 1996). These flow channels are ionospheric signatures of magnetic reconnection events (FTEs). The ionospheric signatures of the coupling include pulsed ionospheric flows (PIFs) in the cusp, which have been observed by HF radars (Walker et al., 1986; Prikryl et al., 1998; McWilliams et al., 2000). These PIFs can be modulated by solar wind Alfvén waves (Prikryl et al., 1999; 2002). Solar wind Alfvén waves (Belcher and Davis, 1971) that couple to the magnetosphere-ionosphere system are associated with high-intensity long-duration continuous auroral activity (HILDCAA) (Tsurutani and Gonzalez, 1987; Tsurutani et al., 1990). Spacecraft observations of the polar cap and auroral zone noted auroral patches during HILDCAA intervals due to the southward component of Alfvén waves causing reconnection (Guarnieri 2006). The durations of the southward component of Alfvén waves influence the geo-effectiveness, and the substorm and magnetic storm developments. However, in this paper we focus on the immediate dayside ionospheric response to the IMF during the impact of corotating interaction regions at the leading edge of high-speed streams.

The ionospheric currents, including auroral electrojets, have long been recognized as sources of atmospheric gravity waves (AGWs) (Chimonas and Hines, 1970) propagating globally in the neutral atmosphere (Richmond, 1978; Hunsucker, 1982; Mayr et al., 1984a; 1990; 2013). The AGWs have been observed as traveling ionospheric disturbances (TIDs) in both the dayside and nightside ionosphere using various techniques, including HF radars, ionosondes, and GPS Total Electron Content (TEC) measurements (Hunsucker, 1982; Crowley and Williams, 1987; Crowley and McCrea, 1988; Samson et al., 1989; Bristow and Greenwald, 1996; Afraimovich et al., 2000; Hayashi et al., 2010; Cherniak and Zakharenkova, 2018; Nishitani et al., 2019). Large-scale TIDs (LSTIDs) generally propagate at speeds between 400 and 1,000 ms−1, have wavelengths greater than 1000 km, and periods of 30 - 180 min, while medium-scale TIDs (MSTIDs) tend to propagate at

speeds of 250 - 1,000 ms−1, and have wavelengths of several hundred kilometers and periods of 15 - 60 min (Francis, 1975; Hunsucker, 1982; Zhang et al., 2019).

On the dayside, in addition to polar cap patches, the generation of AGWs can be pulsed by the solar wind Alfvén waves (Prikryl et al., 2005; 2019). In this paper, we present a case study of polar cap patches and TIDs generated in the dayside ionosphere during minor to moderate geomagnetic storms.

## 2 Data sources and methods

The Resolute Bay Incoherent Scatter Radars (RISR) covering latitudes from 75° to 81°N (RISR-N) and from 69° to 75°N (RISR-C) are located at a geographic latitude of 74.70°N and geographic longitude of 94.83°W. The electronically steerable phased array radars, which are effectively capable of sampling in multiple beam directions simultaneously, measure electron density, electron and ion temperature, and flow velocities in the cusp and polar cap ionosphere (Gillies et al., 2016; 2018).

The Super Dual Auroral Radar Network (SuperDARN) (Chisham et al., 2007; Nishitani et al., 2019) is used to measure the line-of-sight (LoS) velocities, to map ionospheric convection, and observe TIDs in the ground scatter (vt.superdarn.org). Ground-based magnetometers from the Geophysical Institute Magnetometer Array (GIMA) (www.asf.alaska.edu/magnetometer/), Geomagnetic Laboratory of the Natural Resources Canada (NRCan) (www.spaceweather.ca), and the Canadian Array for Realtime Investigations of Magnetic Activity (CARISMA) (www.carisma.ca/) are used to identify ionospheric currents as sources of AGWs. The magnetometer data were also accessed through SuperMAG (supermag.jhuapl.edu/mag) (Gjerloev, 2012) and INTERMAGNET (www.intermagnet.org).

The horizontal equivalent ionospheric currents (EICs) and vertical current amplitudes are estimated using the spherical elementary current system (SECS) inversion technique. We applied the SECS inversion technique (Amm and Viljanen, 1999; Weygand, 2009; Weygand et al., 2009; Weygand et al., 2011) to obtain horizontal EICs and vertical current amplitudes from 11 different arrays of ground magnetometers in the North American sector with stations in western Greenland included. Following Weygand et al. (2011), for each of these stations the quiet-time background is subtracted from the measured field to give the disturbance component which determines the EICs.

The Canadian High Arctic Ionospheric Network (CHAIN) (Jayachandran et al., 2009) consists of ionosondes and GPS Ionospheric Scintillation and TEC Monitors (GISTMs) that are configured to record the power and phase of the L1 frequency (1575.42 MHz) and L2 frequency (1227.6 MHz) signals. In this study, both scintillation indices (S4 and sigma phi) and TEC are used. The TEC used in this study is determined using the phase leveling and cycle slip correction method

outlined in Themens et al. (2013), with satellite biases acquired from the Center for Orbit Determination in Europe (CODE, ftp://ftp.aiub.unibe.ch/) and receiver biases determined as detailed in Themens et al. (2015).

In addition to the CHAIN data, Madrigal Line-of-Sight (LoS) TEC data was also gathered from http://cedar.openmadrigal.org/ and used in this study to examine large scale TEC variations over North America. To characterize the TID structures using this data, LoS TEC data from each satellite-receiver pair was detrended by first projecting the LoS TEC to vertical TEC (vTEC) and removing the sliding 60-minute average. For more details on this method, full details can be found in Themens et al. (2022). The vTEC anomalies are mapped along the SuperDARN radar
beams to be compared with the TIDs observed in the mapped ground scatter (Bristow et al., 1994).

The solar wind data are obtained from the Goddard Space Flight Center Space Physics Data Facility (https://spdf.gsfc.nasa.gov/index.html) and the National Space Science Data Center OMNIWeb (http://omniweb.gsfc.nasa.gov) (King and Papitashvili, 2005). Specifically, the interplanetary magnetic field (IMF) data
obtained by ACE (Smith et al., 1999) and Geotail spacecraft (Kokubun et al., 1994) are used.

## 3 Generation of polar cap patches and traveling ionospheric disturbances modulated by solar wind Alfvén waves

Solar wind high-speed streams (HSSs) are permeated with solar wind Alfvén waves (Belcher and Davis, 1971) known to
cause substorms and geomagnetic storms (Tsurutani et al. 1990; 2006), particularly when associated with significant southward IMF $B_z$. Figs. 1a and 1b show the hourly OMNI data of solar wind plasma variables and the geomagnetic storm *Dst* index for two periods in 2016 and 2018. Arrivals of corotating interaction regions (CIRs) at the leading edge of HSSs on March 6 and 14-15, 2016, and on May 5, 2018, triggered moderate to minor geomagnetic storms (Gonzalez et al., 1994) with the *Dst* index reaching maximum negative values of −110, −60 and −64 nT, respectively. The HSS/CIRs were closely
preceded by heliospheric current sheets (HCS) (Smith et al., 1978) that are associated with high-density plasma leading to the magnetic field compression (Smith and Wolfe, 1986; Tsurutani et al. 1995a). Solar wind Alfvén waves are characterized by the Walén relation between velocity *V* and magnetic field *B* (e.g., Yang et al., 2020; and references therein). The corresponding components of the IMF fluctuations and solar wind velocity are (anti)correlated (Prikryl et al., 2002) and this was the case for the events studied here using observations by the ACE and Geotail spacecraft in the upstream solar wind.
While the Alfvén waves are also observed in the high-speed stream proper following the CIRs, the ionospheric response that is the subject of this paper is limited to CIRs where the Alfvén wave amplitudes are higher due to the compression (Tsurutani et al., 1995b), which likely plays a role in the magnetic reconnection at the dayside magnetosphere resulting in polar cap patches and TIDs.

### 3.1 Ionospheric signatures of flux transfer events and polar cap patches

The RISR measurements of electron density, $N_e$, and flow velocities, $V_e$, in the cusp and polar cap ionosphere are used to study four events of the solar wind Alfvén waves coupling to the dayside magnetopause generating polar cap patches. This is supported by observations of HF radar line of sight (LoS) ionospheric velocities and convection maps by SuperDARN, of polar patches over the CHAIN ionosondes, and of GPS scintillation receivers measuring phase variation (scintillation) index $\sigma_\Phi$. The horizontal equivalent ionospheric currents (EICs) that are estimated from the ground-magnetometer data using the SECS inversion technique provide a broader context of the coupling process to ionospheric currents, including the auroral electrojets.

### 3.1.1 Event of March 6, 2016

Fig. 2a shows $N_e$ and anti-sunward $V_e$ averaged over the longitude span of the RISR-N beams (from 75° to 100°W) and RISR-C beams (from 93° to 107°W) (Gillies et al., 2016; see, their Fig. 1) and altitudes between 250 and 450 km. The poleward propagating enhancements in $N_e$ are due to polar cap patches entering and exiting the RISR field of view (FoV). The time series of the ACE IMF $B_y$ and $B_z$ are superposed, time shifted for the best correlation (correspondence) with patches and anti-sunward flows, respectively, to approximately account for the propagation delay between the spacecraft and the ionosphere. This indicates that the duskward deflections of the predominantly dawnward IMF $B_y$ (<0) resulted in a series of poleward convecting polar patches that are correlated with the IMF $B_y$. The southward IMF was followed by anti-sunward flows that were diminished or stopped when the IMF $B_z$ switched back to northward. While the sustained southward IMF $B_z$ is the condition for the continuous (quasi-steady) magnetic reconnection it is the impulsive reconnection that leads to formation of polar cap patches.

The first few patches (enhancements in $N_e$) started to be observed by RISR-N north of 75°N between 16:00 and 17:00 UT and were not detected by RISR-C (Fig. 2a). This implies that the cusp was in the RISR-C FoV since polar patches are known to be produced by flow channels in the cusp. The very first density patch starting at ~16:00 UT followed the first IMF $B_y$ duskward deflection after the onset of anti-sunward flows due to the southward $B_z$. The FTE signatures of the impulsive magnetic reconnection at the dayside magnetopause were observed by RISR-C in the cusp ionosphere.

Fig. 3 shows the ionospheric currents (EICs) at 110 km mapped in geographic coordinates with the latitudinal maxima of EICs at each longitude grid, highlighted in bold, indicating the locations of westward and eastward electrojets. The flow vectors measured by RISR are coded in color. The GPS ionospheric pierce points (IPPs) at 110 km are shown as circles scaled by the CHAIN GPS phase variation values, $\sigma_\Phi$. It has been shown (Prikryl et al., 2016; 2021a) that, in the auroral zone, IPPs of strong GPS phase scintillation are largely collocated with the electrojet currents. During the period between

15:40 and 17:40 UT of anti-sunward flows driven by the southward $B_z$ before it reversed to northward (Fig. 2a), frequent transient azimuthal westward flows were observed by RISR-C in the cusp that were associated with the IMF $B_y$ duskward deflections (Fig. 2a). The first two azimuthal flow channels intensified and faded between 15:47 to 15:51 UT (Fig. 3a) and between 15:54 and 16:01 UT. More transient azimuthal flows occurred during periods of 16:14-16:16 UT, 16:29-16:39 UT, 16:48-17:00 UT, 17:07-17:12 UT (black rectangles in Fig. 2a) that are associated with duskward deflections of the time-shifted IMF $B_y$ at ~16:15, 16:35, 16:50, and 17:10 UT (black bars above the rectangles in Fig. 2a). These transient azimuthal flows, some of which are shown in Fig. 3, occurred poleward of the CRB identified in the EICs that show a reversal of currents just equatorward of the flow transients at geographic latitude of ~68°N. In some cases when there was ionospheric backscatter the CRB is detectable in the LoS velocities observed by the Kapuskasing radar (not shown).

The large-amplitude swing of the IMF $B_z$ northward stopped the anti-sunward flow for about 30 min (Fig. 2a). After the IMF $B_z$ reversed back to southward the anti-sunward flow was restored and intensified. As the cusp shifted further equatorward after the steep southward reversal of the IMF, azimuthal flow channels in the cusp were not observed any longer by RISR that continued to observe anti-sunward flows and poleward convecting density patches. The patches that followed the duskward IMF $B_y$ deflections must have been produced in the cusp south of RISR. Although the SuperDARN radars in Kapuskasing (KAP) and Saskatoon (SAS; operating in a special mode) observed some of the anti-sunward flows the FTE signatures of transient azimuthal flows could not be identified because of insufficient ionospheric backscatter, which was often mixed with ground scatter.

RISR observed copious density patches propagating in the polar cap. The SuperDARN global ionospheric convection map (Fig. 4a) shows an expanded convection zone with intense flows from the dayside portions of the dawn and dusk cells through the cusp and into the polar cap, where the anti-sunward flows and patches were observed by RISR. The corresponding GPS TEC map (Fig. 4b), as a function of the Altitude Adjusted Corrected Geomagnetic (AACGM) latitude and magnetic local time (MLT), shows a TOI broken into patches. For the GPS phase variation values $\sigma_\Phi > 0.1$ rad, the IPPs at 350-km altitude that are superposed on the TEC map are collocated with the TOI fragmented into patches (further discussed in Section 3.3).

**3.1.2 Events of March 14-15, 2016**

Following the arrival of HSS/CIR that caused a minor geomagnetic storm (Fig. 1a) a series of polar cap patches were generated by solar wind Alfvén waves coupling to the dayside magnetosphere on March 14 and 15. Anti-sunward flows and density patches convecting poleward were observed by RISR (Fig. 2b and 2c). Similar to the event discussed in the previous

section, the patches are approximately correlated with the IMF $B_y$ duskward deflections of the time-shifted and predominantly dawnward IMF $B_y$ (<0) observed by the Geotail spacecraft passing in front of the subsolar bow shock.

On March 14, the main difference is that the IMF $B_z$ remained predominantly northward (>0) but underwent frequent reversals to southward. The large northward $B_z$ before 19:10 UT resulted in a reduced ionospheric convection zone that was, due to duskward $B_y$ (>0), dominated by the dusk convection cell, but also a reverse (two-)cell convection in the noon sector (see, e.g., Gosling et al., 1990; Crooker et al., 1992; Provan et al., 2005; Lu et al., 2011). With the reverse convection (lobe) cell over Resolute Bay, RISR observed sunward and duskward flows, a signature of the lobe reconnection. At this time, the RISR observed electron densities, which were not segmented into poleward propagating patches.

At ~19:10 UT the flows reversed from sunward to anti-sunward following the IMF $B_z$ reversal to southward that resulted in a magnetic reconnection pulse at the dayside magnetopause and the convection settled into a regular two-cell pattern. At the same time, as the IMF $B_y$ was reversing from duskward (>0) to dawnward, it oscillated and rebounded briefly duskward. At 19:10 UT, RISR observed a complex mixture of dawnward and duskward (westward and eastward) transient azimuthal flows (Fig. 5a) what could be a mixture of lobe and subsolar FTE signatures in the cusp ionosphere. This was followed by anti-sunward flows and by a density patch that convected poleward in the RISR-N FoV (Fig. 2b). Another azimuthal flow channel occurred at 19:22-19:23 UT (Fig. 5b) but the IMF $B_z$ reversed to northward and the anti-sunward flows ceased.

After the IMF turned southward for the second time, the anti-sunward flows were restored, a duskward deflection of the IMF $B_y$ followed, a strong azimuthal flow channel was observed by RISR-C (Fig. 5c) and a density patch convected poleward with the antisunward flow (Fig. 2b). Following the next southward $B_z$ dip accompanied by a duskward deflection of $B_y$ RISR-C observed intense azimuthal flows, RISR-N observed enhanced anti-sunward flows (Fig. 5d), and the next density patch convected poleward just after 20:00 UT (Fig. 2b). Although the propagation delay used in Fig. 2 is only approximate and may vary with time, the same arguments can be applied to density patches observed after 21:00 UT except that the cusp moved south of the RISR FoV. The anti-sunward flows appear to have intensified with each southward swing of the IMF $B_z$ associated with a duskward deflection of $B_y$, and the patches were produced.

The southward $B_z$ is known to drive pulsed ionospheric flows (PIFs) (McWilliams et al., 2000). Starting with the first IMF southward turning, PIFs were observed by the SuperDARN Saskatoon (SAS) radar in the LoS ionospheric velocities (negative is away from the radar) (Fig. 6). Even though the ionospheric backscatter was relatively sparse there is an approximate correspondence between the southward swings of the IMF $B_z$ and PIFs.

Polar cap patches observed by RISR on March 15 are also approximately correlated with the IMF $B_y$ duskward deflections of the time-shifted and predominantly dawnward IMF $B_y$ (<0) observed by Geotail spacecraft (Fig. 2c) that was located in front

of the post-noon bow shock. The anti-sunward flows started at ~18:20 UT following the southward IMF $B_z$ and continued for another 3 hours until RISR moved out from the convection throat and deeper into the post noon sector. Following the southward $B_z$ and duskward $B_y$ turning at ~18:10 UT the cusp rapidly shifted equatorward of the RISR FoV, a brief transient azimuthal flow was observed (not shown), and the first density patch convected poleward and reaching 75.5°N latitude at 18:30 UT (Fig. 2c). The production of polar patches was inhibited, or reduced, when $B_z$ become positive or near zero.

The density patches that followed were produced while the cusp was equatorward of RISR. At ~19:30 UT when the IMF $B_z$ reached zero and become mildly positive the cusp re-entered the RISR-C FoV. The next weak patch that was not observed by RISR-C started to be seen by RISR-N (75.5°N the cusp) at 19:45 UT. With the IMF $B_z$ fluctuating around zero and following the duskward $B_y$ deflection a brief transient azimuthal flow burst was observed at 19:40 UT (not shown). The rest of the density patches after 20:00 UT all started to be observed at 70°N when the cusp again moved equatorward, or to the southern edge, of the RISR FoV. The latter was likely the case for the dense patch observed starting at ~20:40 UT following the onset of anti-sunward flow associated with the southward $B_z$ when an azimuthal eastward flow channel was observed in the southern edge of RISR (not shown).

### 3.1.3 Event of May 5, 2018

Similar to the March 6, 2016, fast anti-sunward flows but weaker polar cap patches (Fig. 2d) were approximately correlated with the time-shifted large amplitude alfvénic fluctuations of the IMF $B_z$ and $B_y$ components, respectively. After the IMF $B_y$ reversed from duskward (>0) it remained predominantly dawnward (<0) but was undergoing frequent duskward deflections. The large-amplitude southward turnings of the IMF were followed by anti-sunward flows that were reversed or diminished when the IMF $B_z$ switched back to northward. In the cusp, a series of PIFs were observed by the Kapuskasing radar. Fig. 7 shows the line-of-sight ionospheric velocities with the ground scatter coded in grey color. Despite the ground scatter interference and sparse ionospheric backscatter an approximate correspondence can be made between the time-shifted IMF $B_y$ and PIFs prior to 19:00 UT that is similar to the correlation with polar cap patches observed by RISR (Fig. 2d). Most of the PIFs appear to have been driven by the duskward IMF $B_y$ deflections. This is consistent with the transient flows observed by Provan et al. (1998), except in their case the IMF $B_y$ was predominantly duskward (>0).

Before 17:00 UT, when the polar cap patches were observed by RISR-N, the cusp was in the RISR-C FoV. The first poleward propagating weak density enhancement appeared in the RISR-N FoV (75.5°N) at ~14:50 UT following the onset of anti-sunward flow due to the southward $B_z$ and the first duskward deflection of the IMF $B_y$ before it remained predominantly dawnward (Fig. 2d). More density patches started to be observed from 16:00 UT. While the cusp was in the RISR-C FoV transient azimuthal flows occurred during periods of 14:49-14:56 UT, 15:39-15:50 UT, 15:55-16:01 UT, 16:09-16:11 UT, 16:28-16:29 UT, 16:35-16:36 UT, 16:54, 17:07, and 17:25-17:27 UT (black rectangles in Fig. 2d) that are associated with

duskward deflections of the time-shifted IMF $B_y$ at ~14:47, 15:46, 16:00, 16:11, 16:29, 16:35, and 17:00 UT (black bars above the rectangles in Fig. 2d). Fig. 8 shows four examples of azimuthal transient flows that occurred poleward of the CRB, which is consistent with observations of FTE signatures (Provan et al., 1998).

Following the intensification of the anti-sunward flow caused by the southward $B_z$ turning at 17:50 UT the highest density
patch convected across the whole RISR FoV starting just before 18:00 UT (Fig. 2d). The cusp has shifted equatorward and RISR could not observe azimuthal flows, but a narrow PIF was observed by the Kapuskasing radar at lower latitudes (Fig. 7). While most of the density patches were associated with a significant southward IMF $B_z$ the next patch was observed between 19:10 and 20:00 UT (Fig. 2d) when the IMF $B_z$ fluctuated about zero. Nevertheless, brief azimuthal flows were observed at the southern edge of RISR-C FoV at 19:04-19:05 UT (not shown).
In comparison with the March events the electron densities of the patches were lower and did not result in any significant GPS phase variation $\sigma_\Phi$, which is discussed in Section 3.3.

### 3.2 Traveling ionospheric disturbances

The electric fields that drive the $\mathbf{E} \times \mathbf{B}$ ionospheric convection in the F region map to the E-region driving ionospheric currents, including auroral electrojets. The ionospheric currents are sources of Joule heating in the lower thermosphere
launching atmospheric gravity waves (AGWs), which in turn cause traveling ionospheric disturbances (TIDs). In this section, we examine the observations of AGWs/TIDs generated by solar wind Alfvén waves coupling to the dayside magnetosphere during the four events discussed above. Figs. 9a-d show TIDs observed in the detrended vertical vTEC and the radar ground-scatter power focused and defocused by TIDs moving equatorward. In the case of LSTIDs with a wavelength of more than 1000 km the tilted isopycnic surfaces divert the refracted radio waves back and forth thus
modulating the range of the ground scatter. The time-shifted time series of the IMF $B_z$ observed by ACE or Geotail are superposed.

### 3.2.1 TIDs on March 6, 2016

Fig. 9a shows TIDs observed in the ground scatter power and in the vTEC anomaly mapped at range gates along the
Christmas Valley West (CVW) radar beam 12 (Fig. 10a) looking northwest over western Canada on March 6, 2016. Rather than showing the ground scatter slant range, the ground scatter range mapping discussed by Bristow et al. (1994) and Frissell et al. (2014) is applied to reflect the TID location in the ionosphere. The IMF $B_z$, the X component of the ground magnetic field measured in Baker Lake (BLC), and time series of the latitudinal maxima in EICs at the longitude of 99.6°W, are superposed. As the fluctuating IMF started to turn southward fluctuating ionospheric currents sensed by magnetometers
launched AGWs.

While the HF propagation may introduce some uncertainty to the radar observations of TIDs there is a one-to-one correspondence between the BLC X-component fluctuations due to westward currents inverted to EICs and the TIDs (marked 1 to 5 in Fig. 9a). After ~19:00 UT it was the eastward electrojet intensifications that launched TIDs (marked 6 to 9). Along the radar beam at near ranges, there is an approximate one-to-one correspondence between the vTEC anomaly variations due to TIDs that propagated to midlatitudes and the TIDs observed in the ground scatter, although TIDs 1 and 2 are not resolved in the vTEC anomaly. At far ranges, MSTIDs and LSTIDs before and after ~19:00 UT, respectively, are observed propagating from their sources, the ionospheric currents at high latitudes. The LSTIDs are correlated with the long-period fluctuations of the IMF $B_z$, while the MSTIDs appear to be consistent with shorter period IMF fluctuations.

Figs. 3c and 3d were selected to show transient azimuthal flows observed by RISR at 16:30 and 16:52 UT. These times approximately correspond to the times of EICs maxima (Fig. 9a) due to the westward electrojet spanning from Alaska across the northern Canada that likely launched TIDs 1 and 2. Because of the limited coverage by GPS receivers at high latitudes the MSTIDs could be detected only over Alaska/Yukon, but LSTIDs were observed at midlatitudes after ~19:00 UT. Fig. 11a shows intense westward and eastward electrojets that launched the LSTID 5 (Fig. 9a) that was observed in the detrended vTEC maps an hour later (Fig. 11b). The LSTID 7 that was launched at ~20:00 UT by an intensification of eastward electrojet (Fig. 11c) later propagated to midlatitudes. Fig. 11d shows it stretching from the west coast. The non-propagating enhanced vTEC region at higher latitudes stretching from the central to eastern Canada is due to storm enhanced density discussed in Section 3.3.

The one-to-one correspondence between the IMF $B_z$ fluctuations and auroral currents that launched the TIDs is less evident, but the IMF $B_z$ fluctuations were similar to the X-component measured by another magnetometer further west in Barrow (BRW), Alaska, between ~16:30 and 19:00 UT (not shown). However, the coupling of solar wind ULF waves to the magnetosphere is a complex process that can involve pressure pulses, mode conversion to fast mode waves and field line resonances on closed magnetic field lines (e.g., Prikryl et al., 1998; and references therein). There were large proton densities (up to ~35 cm$^{-3}$) and dynamic pressure fluctuations (up to 13 nPa) observed in the solar wind (not shown) that likely contributed to modulating the ionospheric currents.

**3.2.2 TIDs on March 14-15, 2016**

Fig. 9b shows TIDs (1-9) observed by the CVE radar beam 5 (Fig. 10b) looking northeast over the western and central Canada on March 14, 2016. The detrended vTEC along the beam appears to have detected MSTIDs at ranges between ~500 and 1000 km before ~19:30 UT, and LSTIDs thereafter, but MSTIDs 5-6 and the structure preceding the LSTID 7 that can be seen in the ground scatter are not resolved in the vTEC anomaly.

Superposed time series of the latitudinal maxima in EICs are averages over longitudes between 70° and 110°W. The southward IMF $B_z$ dips of the IMF that modulated the anti-sunward flows and produced polar cap patches (Fig. 2b) also played a role in driving intensifications of ionospheric currents launching AGWs/TIDs. The latitudinal maxima in EICs indicate the initiation (launch) times of AGWs that approximately correspond with TIDs observed by the CVE radar.

There is an approximate correspondence between the southward turning of the IMF $B_z$ and EICs/TIDs 1-6. This becomes less clear after 20:00 UT partly because of westward and eastward electrojet intensifications at different latitudes. The fluctuating EIC that peaked at 20:34 UT due to westward current at high latitudes was associated with a fluctuation of the ground scatter power but the corresponding fluctuation in the vTEC anomaly at close range (~200 km) is poorly resolved. The southward turning of the IMF $B_z$ at ~21:00 UT (Fig. 9b) coincided with the launch of LSTID 7. The next southward turning of the IMF

is associated with a peak in EIC at 21:10 UT but this time due to the eastward electrojet at significantly lower latitudes (Fig. 12a). Beam 5 does not reveal a distinct TID signature, except for the ground scatter power increase. However, another TID (7+) initiation superposed on the TID 7 was observed by beams 0 and 1 (not shown). The LSTID 7+ was later observed in the detrended vTEC maps (Fig. 12b).

The LSTID 8 and 9 were observed by the CVE radar starting at ~22:00 and 22:30 UT. Although two peaks in the averaged eastward EICs are not well resolved the LSTIDs that are associated with southward turnings of the IMF $B_z$ were launched by intensifications of the eastward electrojet (Figs. 12c). The stronger LSTID 9 was later observed overtaking the weaker LSTID 8 ahead of it over the southern U.S. (Figs. 12d).

On March 15, 2016, the CVE radar beam 15 (Fig. 10c) looking northeast across the northwestern U.S. and eastern Canada observed LSTIDs in the ground scatter (Fig. 9c) at far ranges on the second HF propagation hop between the ground and ionosphere, which introduces some uncertainty in the HF propagation. These LSTIDs were launched by intense westward and eastward electrojet currents modulated by long period variations of the IMF $B_z$ observed by Geotail. The time series of the latitudinal maxima in EICs at the longitude of 92.7°W show long period EIC variations that are associated with the

LSTIDs and correspond with the vTEC anomaly variations mapped along the beam. The equatorward propagating LSTIDs were observed in the detrended vTEC maps as they started to appear along the border between the U.S. and Canada. Figs. 13a and 13b show intensifications of the westward and eastward electrojets that launched the LSTIDs observed later in the detrended vTEC maps (Fig. 13c-d).

**3.2.3 TIDs on May 5, 2018**

    Fig. 9d shows TIDs observed by the BKS radar beam 15 (Fig. 10d) looking northwest over the central Canada on May 5, 2018. There is a one-to-one correspondence between the large-amplitude IMF $B_z$ due to solar wind Alfvén waves and TIDs

observed in the ground scatter and the vTEC anomaly mapped along the beam. Also, the peaks in the time series of the maximum EICs at longitude of 134°W are correlated with the southward IMF $B_z$ and the TIDs. While at this longitude the first two latitudinal maxima of EICs associated with TIDs are weak, at longitudes of 148° and 155°W (not shown) these peaks are more pronounced and coincide with the first two TIDs and the time-shifted IMF $B_z$ negative peaks. The peaks of the latitudinal maxima of EICs indicate approximate times when the AGWs that caused the TIDs were launched.

The southward IMF turnings due to large-amplitude Alfvén waves (12:00-18:00 UT) were followed by intensifications of the auroral electrojets each launching a TID (Fig. 9d). The next EIC intensification at ~18:30 UT launched another strong TID. As the Geotail IMF $B_z$ turned from southward to mildly northward and fluctuated, weaker TIDs followed between 20:00 and 21:30 UT. After the IMF turned to southward again strong TIDs were observed again. Figs. 14a and 14b show intensifications of the westward and eastward electrojets that launched the LSTIDs observed later in the detrended vTEC maps (Fig. 14c-d).

### 3.3 GPS phase variation in the cusp, polar cap and auroral oval

The CHAIN GPS phase variation has been linked to polar cap patches, cusp and auroral precipitation/currents (Prikryl et al., 2011; 2016; 2021a). While this is not the focus of the present paper, it is of interest to compare the above events in terms of the temporal and spatial occurrence of the GPS TEC and phase variation.

The strongest polar cap patches that were observed by RISR (Fig. 2a) convected over the CADI ionosondes in Resolute Bay and Eureka. The fixed frequency ionogram from Eureka (Fig. 15a) shows the passing density patches in the zenith over the ionosonde as the U-shaped structures. The calibrated GPS vTEC corrected for biases shows a good correspondence with the patches observed by the ionosonde. The relative variations of the slant TEC resulted in GPS phase variation, $\sigma_\Phi$, reaching up to 1 rad (Fig. 5c). In comparison, on March 14-15, 2016 and May 5, 2018 the patches observed by ionosondes were significantly weaker and caused only weak to moderate GPS phase variations.

Fig. 16a-d shows the percentage occurrence of phase variation, $\sigma_\Phi$, above a given threshold as a function of the Altitude Adjusted Corrected Geomagnetic (AACGM) latitude and magnetic local time (MLT). The percentage occurrence is determined for the grid bins of 0.25 h MLT × 1° AACGM latitude assuming the IPP height of 350 km (Prikryl et al., 2016). The moderate geomagnetic storm on March 6, 2016 (Fig. 16a) resulted in the highest occurrence of GPS phase variation caused by the fast convection of the storm enhanced density (SED) plasma irregularities from the dayside ionosphere through the cusp where the TOI was segmented into polar cap patches. In Fig. 4 discussed in Section 3.1.1, the CHAIN GPS IPPs are superposed on the ionospheric convection and GPS TEC maps as a function of magnetic latitude and MLT. The IPPs shown as circles scaled by the GPS phase variation values, $\sigma_\Phi$, are collocated with the TOI fragmented into patches.

In relation to the auroral electrojets, Figure 17 shows the GPS phase variation occurrence as a function of AAGCM latitude and UT overlaid with contours of the east component of the EIC $J_y$ current averaged over the longitude grids between 86° and 93°W with the EICs transformed to geomagnetic coordinates using the magnetic declination at each grid cell. To conform to the 15 min grid span used for the scintillation occurrence map, the west-to-east $J_y$ current component is averaged over 15 min. Consistent with the previous results (Prikryl et al., 2016) the highest occurrence of GPS phase scintillation in the auroral zone is associated with the westward electrojet and the poleward edge of the eastward electrojet. At high latitudes, the highest occurrence of $\sigma_\Phi > 0.1$ rad is in the cusp and polar cap during times of dense polar cap patches observed by RISR (Fig. 2a).

For specific times the GPS IPPs of enhanced phase variation are shown in Figs. 3 and 11 discussed in the previous sections. The IPPs are found to be collocated with the convection reversal boundary and the westward electrojet (Fig. 3), and the poleward edge of the eastward electrojet (Fig. 11a). At high latitudes, even when fast azimuthal flows were observed by RISR they were not collocated with above-threshold values of $\sigma_\Phi$ (Figs. 3a and 3b). It was only when dense polar cap patches convected over the RISR FoV that IPPs with moderate values of $\sigma_\Phi$ were collocated with the fast anti-sunward flows (Figs. 4).

The minor geomagnetic storms caused significantly less GPS phase variation (Figs. 16c-d). Although large GPS phase variation were collocated with auroral electrojets (Figs. 5 and 14), and some were caused by polar cap patches on March 14-15, 2016, there was very little, or no significant GPS phase variation, associated with the weak polar cap patches in the polar cap on May 5, 2018 (Figs. 8, 16d and 18a). However, even during the latter event Fig. 18a the highest occurrence of GPS phase variation in the auroral zone was associated with the westward electrojet and boundaries between the westward and eastward electrojets. Similar association of the GPS phase variation occurrence with the electrojets is found during the minor geomagnetic storm on March 14-15, 2016 (not shown). The highest occurrence was during the growth phase of the storm (Fig. 16b) when the GPS phase variation was observed mainly in the cusp and polar cap, but also in the post-noon auroral zone (dusk convection cell and SED), nightside auroral zone and possibly a subauroral polarization stream (Prikryl et al., 2016).

## 4 Discussion

The presented multi-instrument observations of polar cap patches in the Canadian Arctic are consistent with previously published results (e.g., Provan et al, 1998) that support the accepted model of polar patch formation (Cowley and Lockwood, 1992). Transient azimuthal flows in the cusp that resulted in the formation of polar cap patches were associated with the IMF $B_y$ fluctuations due to solar wind Alfvén waves. Pulsed ionospheric flows modulated by solar wind Alfvén waves followed by polar cap patches were previously observed (Prikryl et al., 1999; 2002).

The large-amplitude solar wind Alfvén waves in the CIRs at the leading edge of HSSs also modulated the ionospheric currents that were estimated from the ground-based magnetometer data using an inversion technique. The ionospheric currents have been recognized as sources of AGWs causing TIDs. Of course, AGWs/TIDs can be generated by various other sources, including tropospheric weather systems (Bertin et al., 1975, 1978; Waldock and Jones, 1987; Oliver et al., 1997; Nishioka et al. 2013), polar vortex (Frissell et al., 2016), volcanic eruptions, earthquakes, and tsunamis (e.g., Nishitani et al., 2019; Themens et al., 2022), as well as phenomena associated with ion-neutral interactions (Nishitani et al., 2019). However, the case studies of equatorward propagating TIDs observed by SuperDARN and GNSS receivers presented in this paper clearly point to dayside ionospheric currents modulated by solar wind Alfvén waves. This is consistent with the previously published results (Prikryl et al., 2005).

Milan et al. (2017; see, their Fig. 2) reviewed the morphology and dynamics of the electrical current systems of the terrestrial magnetosphere and ionosphere that include DP1, DP2 and DPY currents. The patch formation has been associated with the By-modulated DPY currents (Hall currents associated with FCEs) (Friis-Christensen and Wilhjelm, 1975; Clauer et al., 1995; Stauning et al. 1994, 1995; Prikryl et al. 1999). In the high conductance auroral zone, Hall currents form the eastward and westward auroral electrojets, and the corresponding magnetic perturbations on the ground associated with these Hall currents, are known as the DP1 and DP2 patterns. However, this paper is concerned with the dayside currents, so the TIDs were caused primarily by the DP2 current intensifications.

Finally, it is noted that solar wind coupling to the dayside magnetosphere-ionosphere-atmosphere generating globally propagating gravity waves has been proposed to play a role in the occurrence of extreme weather (Prikryl et al., 2018; 2019, 2021b).

**5 Summary and conclusions**

Production of polar cap density patches and traveling ionospheric disturbances during minor to moderate geomagnetic storms caused by corotating interaction regions at the leading edge of high-speed streams is studied using incoherent scatter radars and networks of HF radars, ionosondes and magnetometers, and GPS receivers. Solar wind Alfvén waves modulated the magnetic reconnection at the dayside magnetopause. The ionospheric signatures of flux transfer events that resulted in formation of polar cap patches were observed in the cusp. The coupling process also modulated the ionospheric convection and the intensity of ionospheric currents, including the auroral electrojets. The horizontal equivalent ionospheric currents were estimated from the ground-based magnetometer data using an inversion technique. Intensifications of ionospheric currents launched atmospheric gravity waves causing traveling ionospheric disturbances that were observed in the HF ground scatter and detected in the detrended GPS vTEC maps. The GPS phase scintillation index obtained by specialized

GPS scintillation receivers was highest during the moderate geomagnetic storm. The GPS phase variation was caused by intense convection of storm enhanced density plasma through the cusp into a dense tongue of ionization segmented into patches. In the auroral zone the highest occurrence of GPS phase variation was collocated with the westward electrojet and boundaries between the westward and eastward electrojets.

*Data availability.* The solar wind data can be obtained from the NSSDC OMNIWeb http://omniweb.gsfc.nasa.gov. The ground-based magnetometer data can be accessed from the Geophysical Institute Magnetometer Array (GIMA) (www.asf.alaska.edu/magnetometer/), Geomagnetic Laboratory of the Natural Resources Canada (NRCan) (www.spaceweather.ca), and the Canadian Array for Realtime Investigations of Magnetic Activity (CARISMA) (www.carisma.ca/), SuperMAG (https://supermag.jhuapl.edu/mag/) and INTERMAGNET (www.intermagnet.org). RISR-C

data are available at http://data.phys.ucalgary.ca/ and https://madrigal.phys.ucalgary.ca/. SuperDARN data are available at https://www.frdr-dfdr.ca/repo/collection/superdarn. Line-of-Sight TEC data were acquired from the Madrigal database http://cedar.openmadrigal.org/. CHAIN GNSS data are available at http://chain.physics.unb.ca/chain/pages/data_download. Equivalent Ionospheric Currents (EICs) derived using the Spherical Elementary Currents Systems (SECS) technique are available through http://vmo.igpp.ucla.edu/data1/SECS/ and https://cdaweb.gsfc.nasa.gov/pub/data/aaa_special-purpose-

datasets/spherical-elementary-and-equivalent-ionospheric-currents-weygand/.

*Author contributions.* PP conceptualized the project, led the formal analysis and investigation, developed the methodology, administered the project, acquired the resources, developed the software, supervised, validated, and visualized the project, wrote the original draft, and reviewed and edited the paper. RGG acquired the resources, developed the software to process and validate the RISR data. DRT acquired the resources, developed the software for detrending GNSS TEC data, processed

the CHAIN and GNSS data, and reviewed and edited the paper. JMW developed and applied the spherical elementary current system (SECS) inversion technique to derive equivalent ionospheric currents. SC contributed to the effort of detrending the GNSS TEC data and accessed the Madrigal data. EGT acquired the resources, developed the software, processed, validated, and visualized SuperDARN data along with GNSS TEC mapping, and reviewed and edited the paper.

*Competing interests.* The authors declare that they have no conflict of interest.

*Acknowledgments.* Infrastructure funding for CHAIN was provided by the Canada Foundation for Innovation and the New Brunswick Innovation Foundation. CHAIN operation is conducted in collaboration with the Canadian Space Agency (CSA). Contributions by the ACE (N. Nees at Bartol Research Institute, D. J. McComas at SWRI), Geotail (S. Kokubun at STELAB Nagoya University), Wind spacecraft teams, the NASA's SPDF/CDAWeb, and the NSSDC OMNIWeb are acknowledged. RISR-C is funded by the Canada Foundation for Innovation and led by the University of Calgary, University of

Saskatchewan, Athabasca University, and SRI International. The authors acknowledge the use of SuperDARN data. SuperDARN is a collection of radars funded by the national scientific funding agencies of Australia, Canada, China, France, Italy, Japan, Norway, South Africa, United Kingdom, and the United States of America. The Christmas Valley SuperDARN radars are maintained and operated by Dartmouth College under support by NSF grant AGS-1934997. Operations of the

Goose Bay, Kapuskasing, and Blackstone SuperDARN radars are supported by the National Science Foundation under award AGS-1935110. The operation of the Saskatoon radar is supported by the Canada Foundation for Innovation, the Canadian Space Agency, and the Province of Saskatchewan. We thank the many different groups operating magnetometer arrays for providing data for this study, including the THEMIS UCLA magnetometer network (Ground-based Imager and Magnetometer Network for Auroral Studies). AUTUMNX magnetometer network is funded through the Canadian Space Agency/Geospace Observatory (GO) Canada program Athabasca University, Centre for Science/Faculty of Science and Technology. The Magnetometer Array for Cusp and Cleft Studies (MACCS) array is supported by US National Science Foundation grant ATM-0827903 to Augsburg College. The Solar and Terrestrial Physics (STEP) magnetometer file storage is at Department of Earth and Planetary Physics, University of Tokyo and maintained by Dr. Kanji Hayashi (hayashi@grl.s.u-tokyo.ac.jp). The McMAC Project is sponsored by the Magnetospheric Physics Program of National Science Foundation through grant AGS-0245139. The ground magnetic stations operated by the Technical University of Denmark, National Space Institute (DTU Space). The Canadian Space Science Data Portal is funded in part by the Canadian Space Agency contract numbers 9 F007-071429 and 9 F007-070993. The Canadian Magnetic Observatory Network (CANMON) is maintained and operated by the Geological Survey of Canada. DRT's contribution to this work is supported in part through CSA grant 21SUSTCHAI and through the United Kingdom Natural Environment Research Council (NERC) EISCAT3D: Fine-scale structuring, scintillation, and electrodynamics (FINESSE) (NE/W003147/1) and DRivers and Impacts of Ionospheric Variability with EISCAT-3D (DRIIVE) (NE/W003368/1) projects. JMW acknowledges NASA grant: 80NSSC18K0570, 80NSSC18K1220, NASA contract: 80GSFC17C0018 (HPDE), NAS5-02099(THEMIS). SC thanks to the National Science Foundation for support under grant AGS-1935110.

*Financial support.* N/A

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

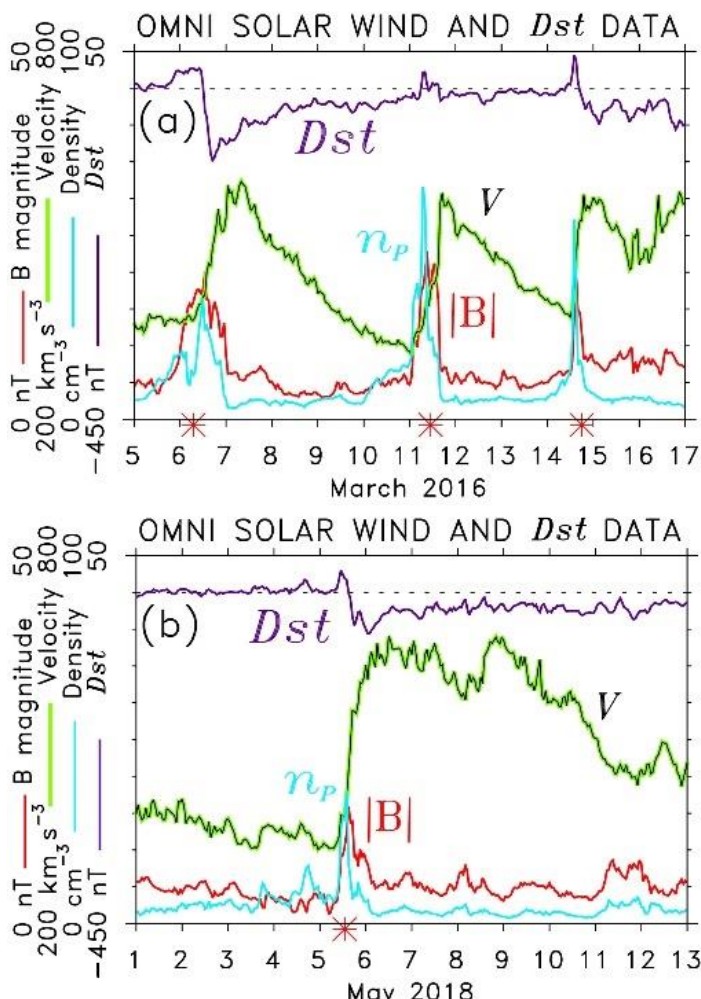

**Figure 1:** The hourly OMNI solar wind velocity, *V*, magnetic field, *B*, proton density, $n_p$, and the *Dst* index for two periods in **(a)** 2016 and **(b)** 2018. Major HSS/CIRs (red asterisks) are shown.

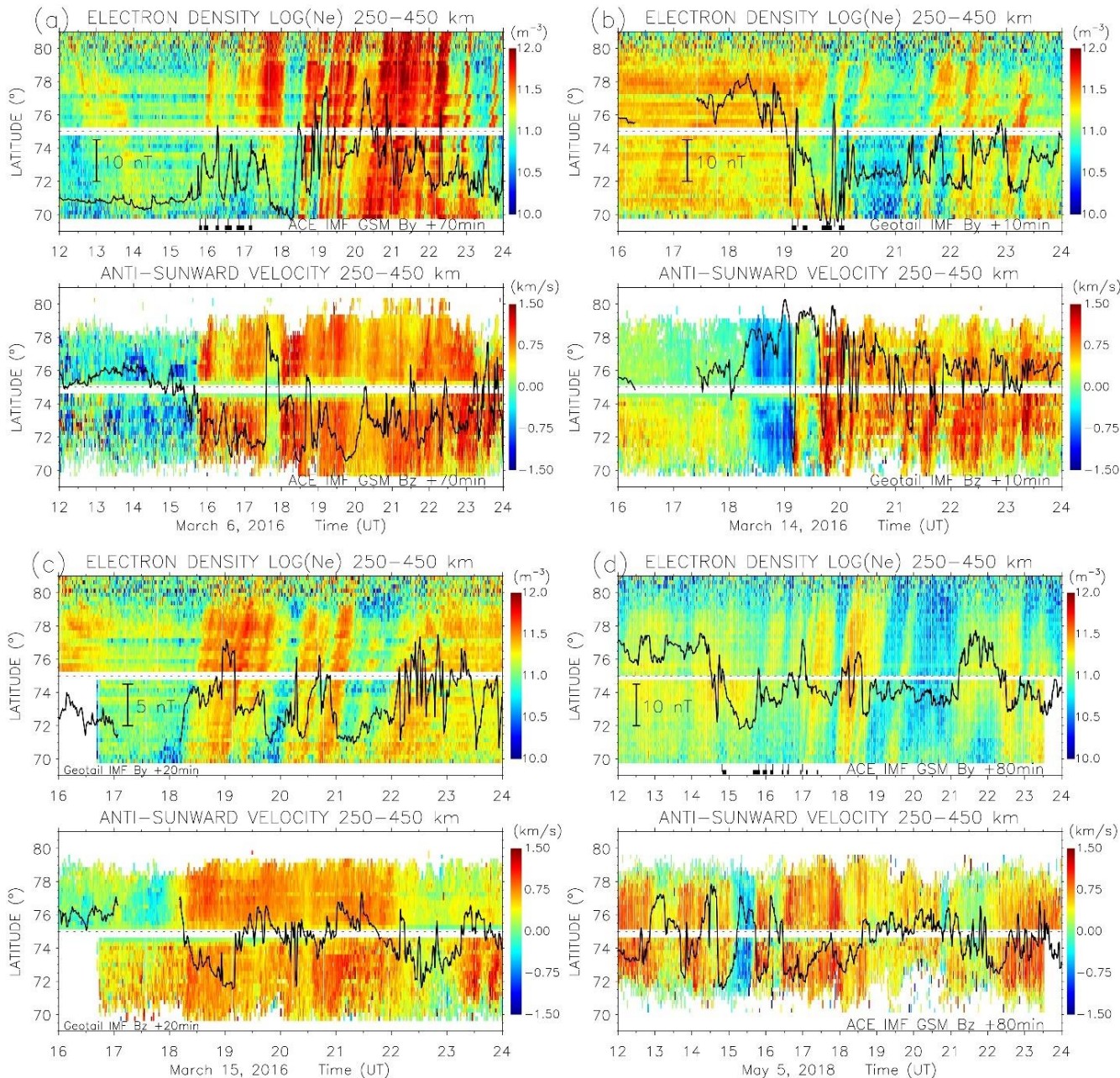

**Figure 2:** The RISR measurements at altitudes between 250 and 450 km of electron density, $N_e$, and anti-sunward flow velocities, $V_e$, on **(a)** March 6, **(b)** March 14, **(c)** March 15, 2016, and **(d)** May 5, 2018. The time-shifted IMF $B_y$ and $B_z$ observed by ACE (GSM) and Geotail (GSE) spacecraft are superposed.

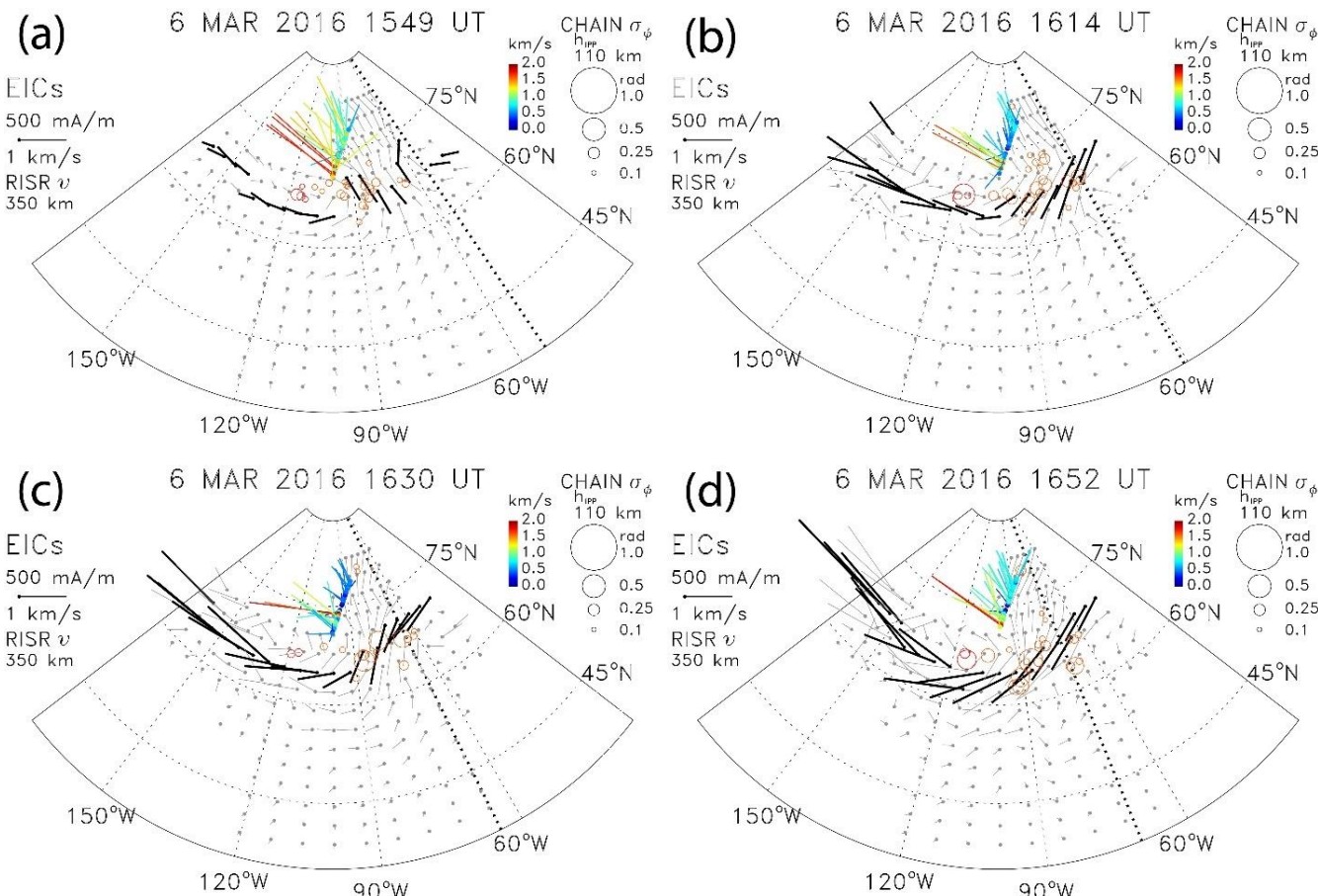

**Figure 3:** The horizontal EICs with their latitudinal maxima at each longitude grid, plasma flows observed by RISR (in color-coded velocities), and the CHAIN GPS IPPs with enhanced phase variation values, $\sigma_\Phi$, (scaled circles). The bold dotted line shows the longitude of local noon.

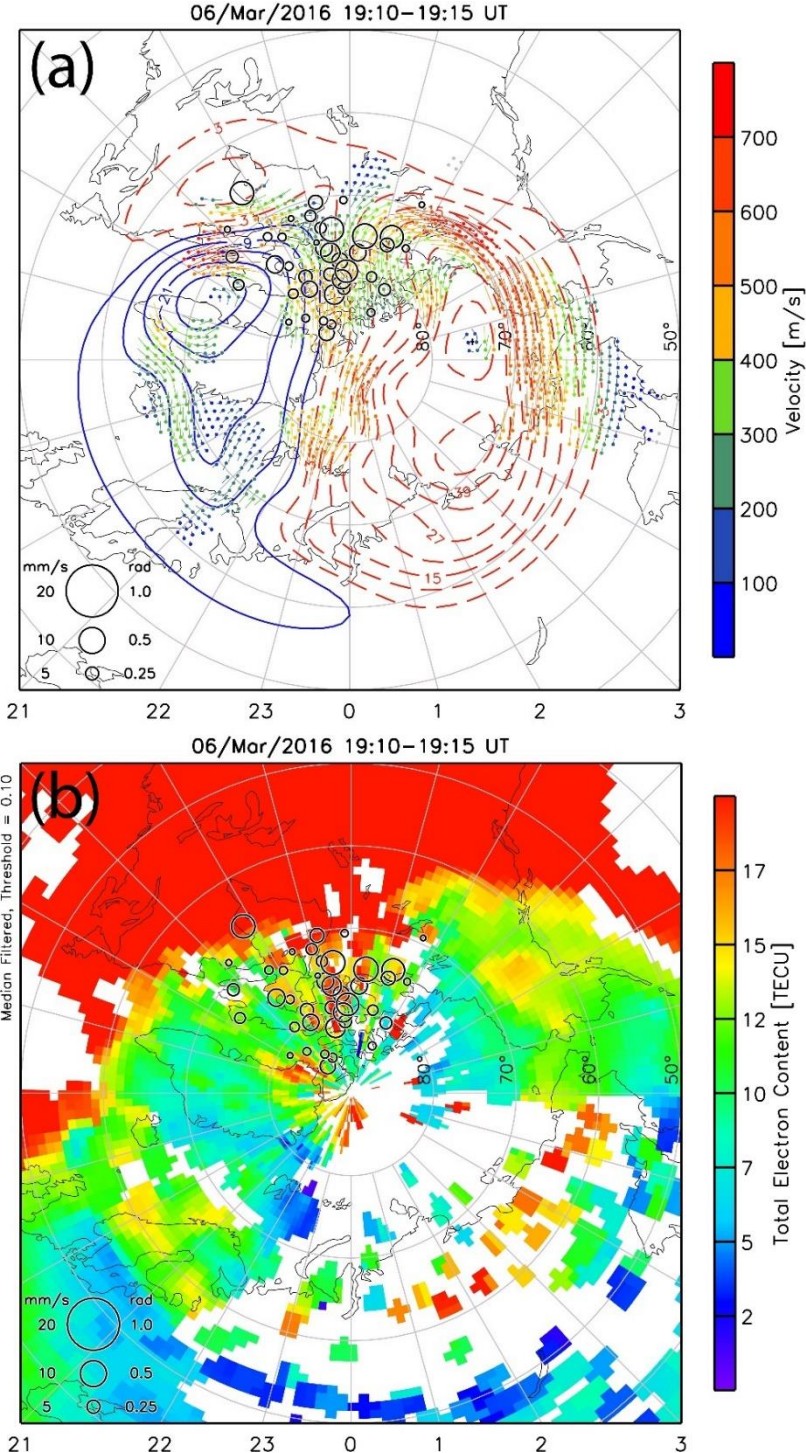

**Figure 4: (a)** The SuperDARN ionospheric convection and potential maps showing expanded convection zone and **(b)** the GPS TEC as a function of magnetic latitude and magnetic local time (MLT). The CHAIN GPS IPPs at 350 km shown as circles scaled by the GPS phase variation values, $\sigma_\Phi$, are superposed.

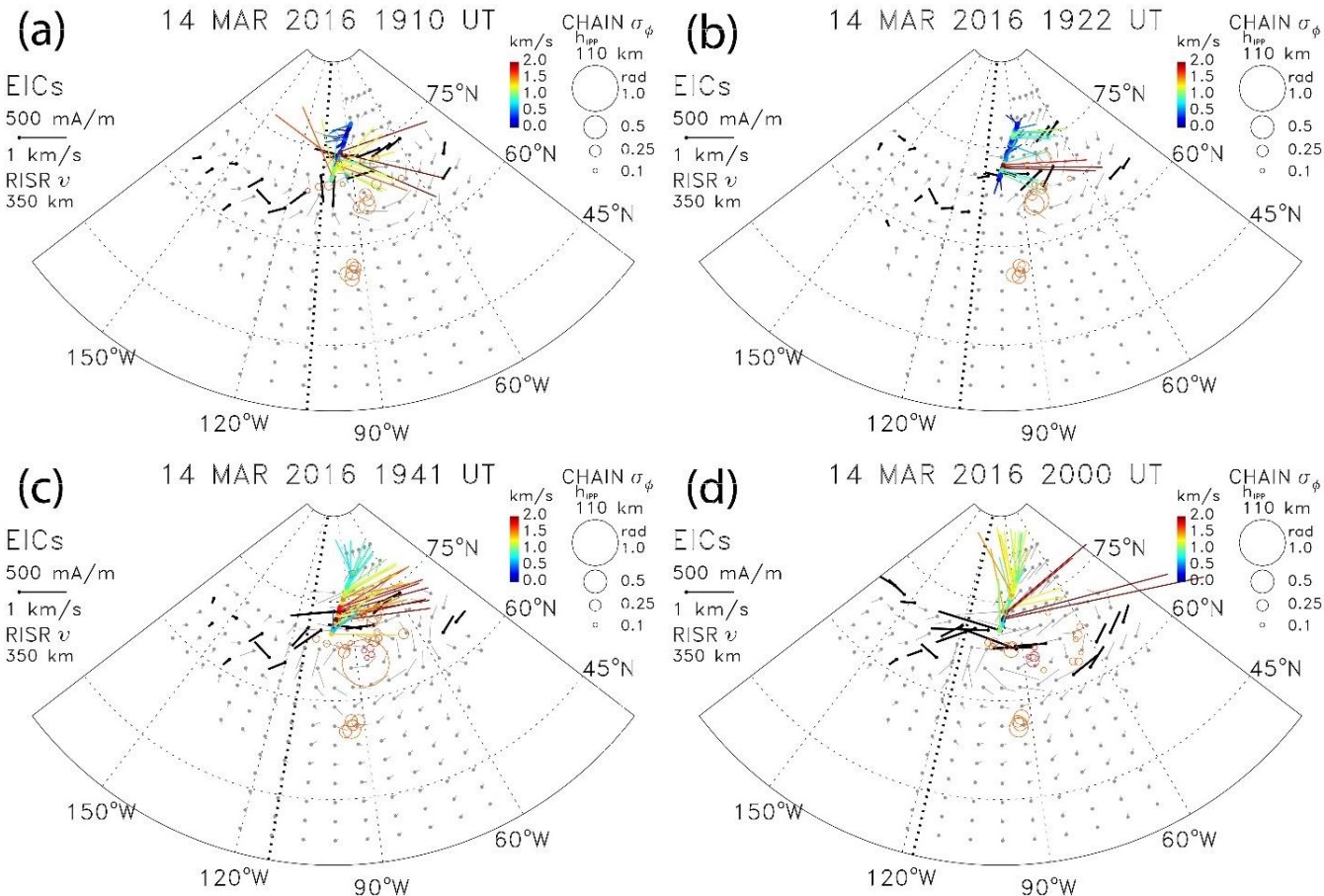

**Figure 5:** The same as Fig. 3 but for March 14, 2016.

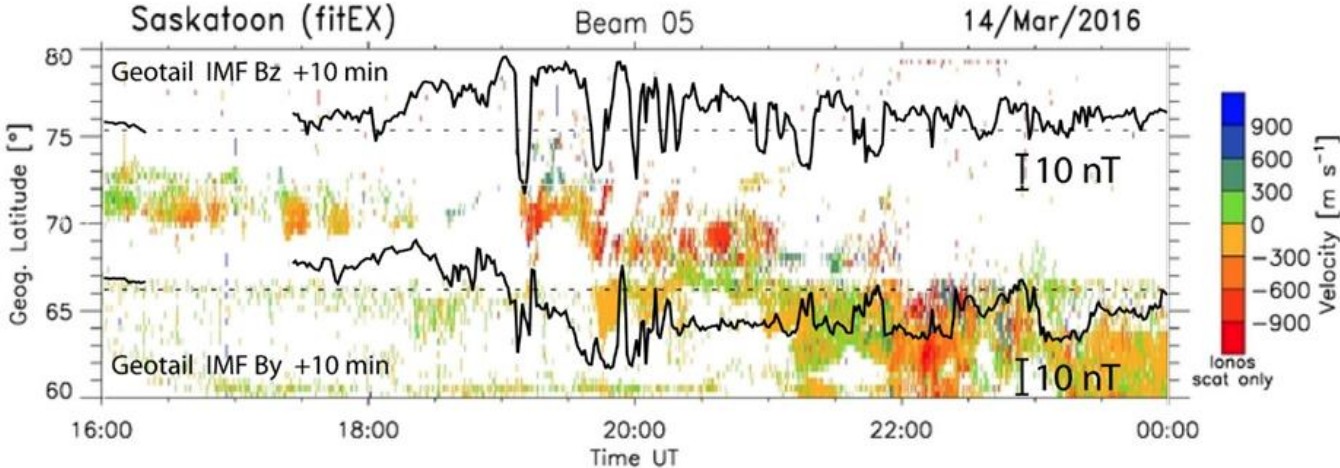

**Figure 6:** The Saskatoon radar line-of-sight velocities as a function of geographic latitude and time (UT). The time-shifted IMF $B_y$ and $B_z$ observed by Geotail are superposed.

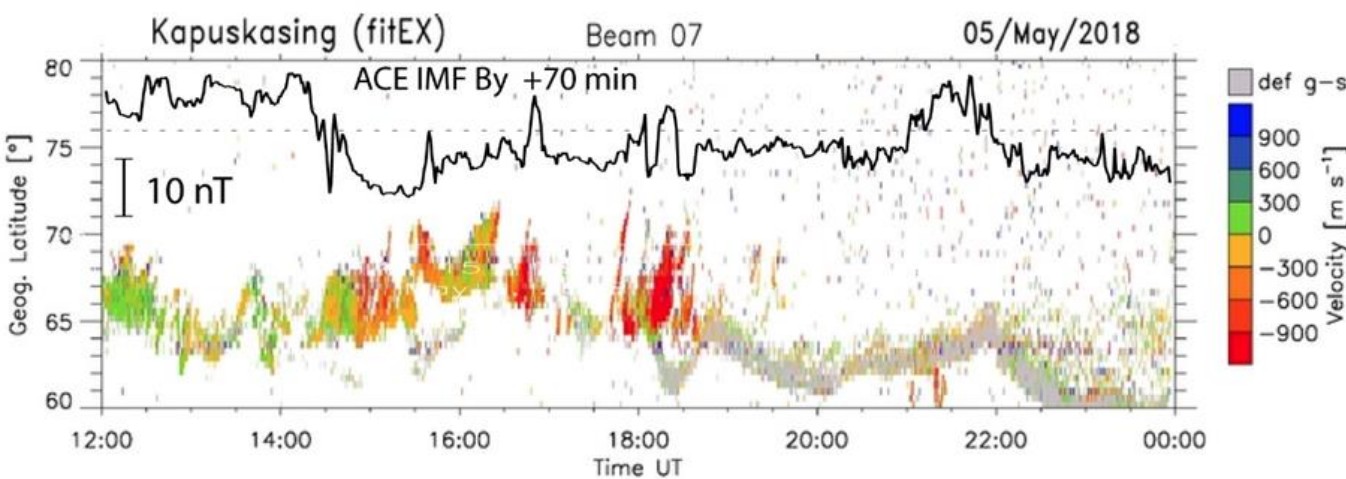

**Figure 7:** The Kapukasing radar line-of-sight velocities as a function of geographic latitude and time (UT). The time-shifted IMF $B_y$ observed by ACE are superposed.

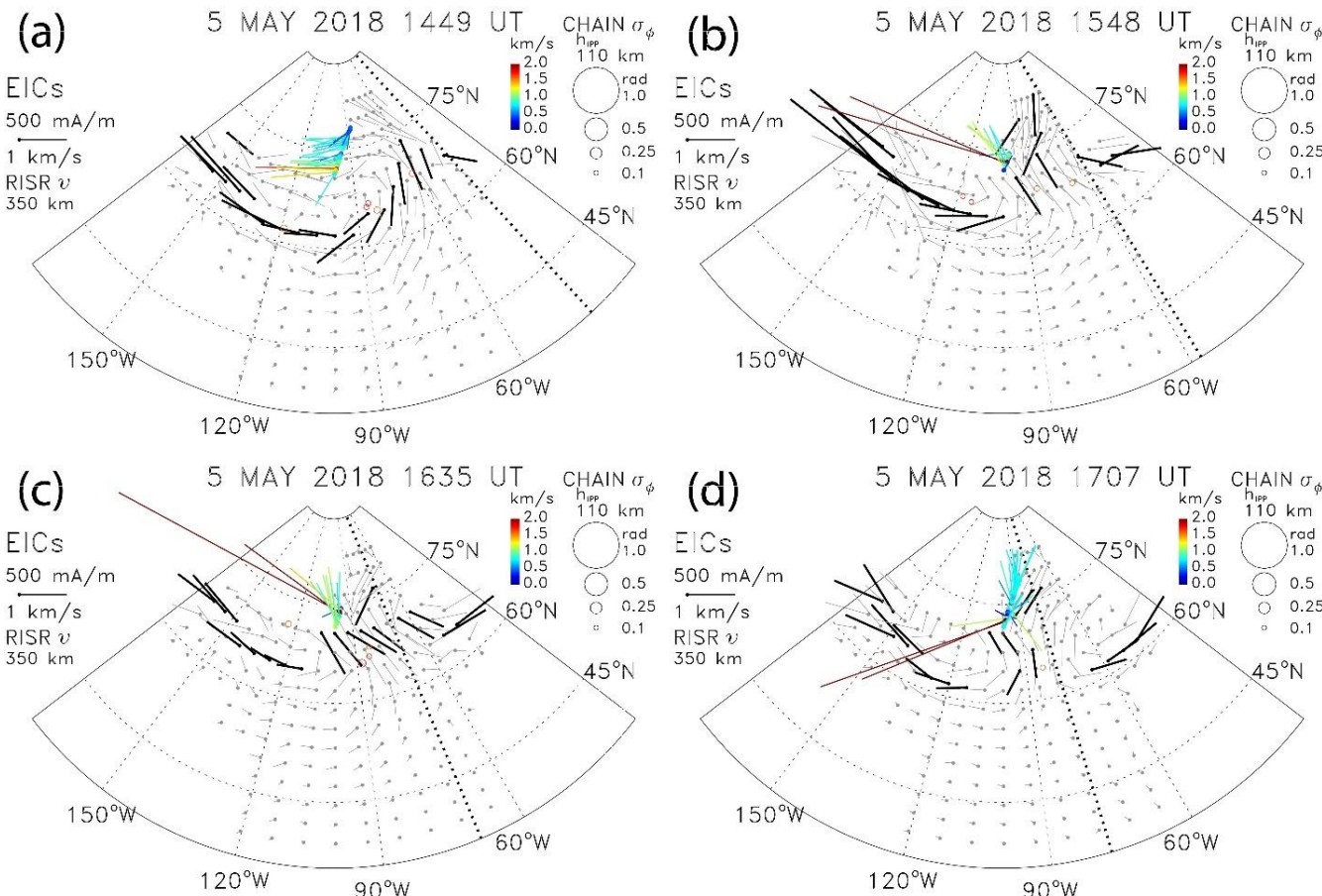

**Figure 8:** The same as Fig. 3 but for May 5, 2018.

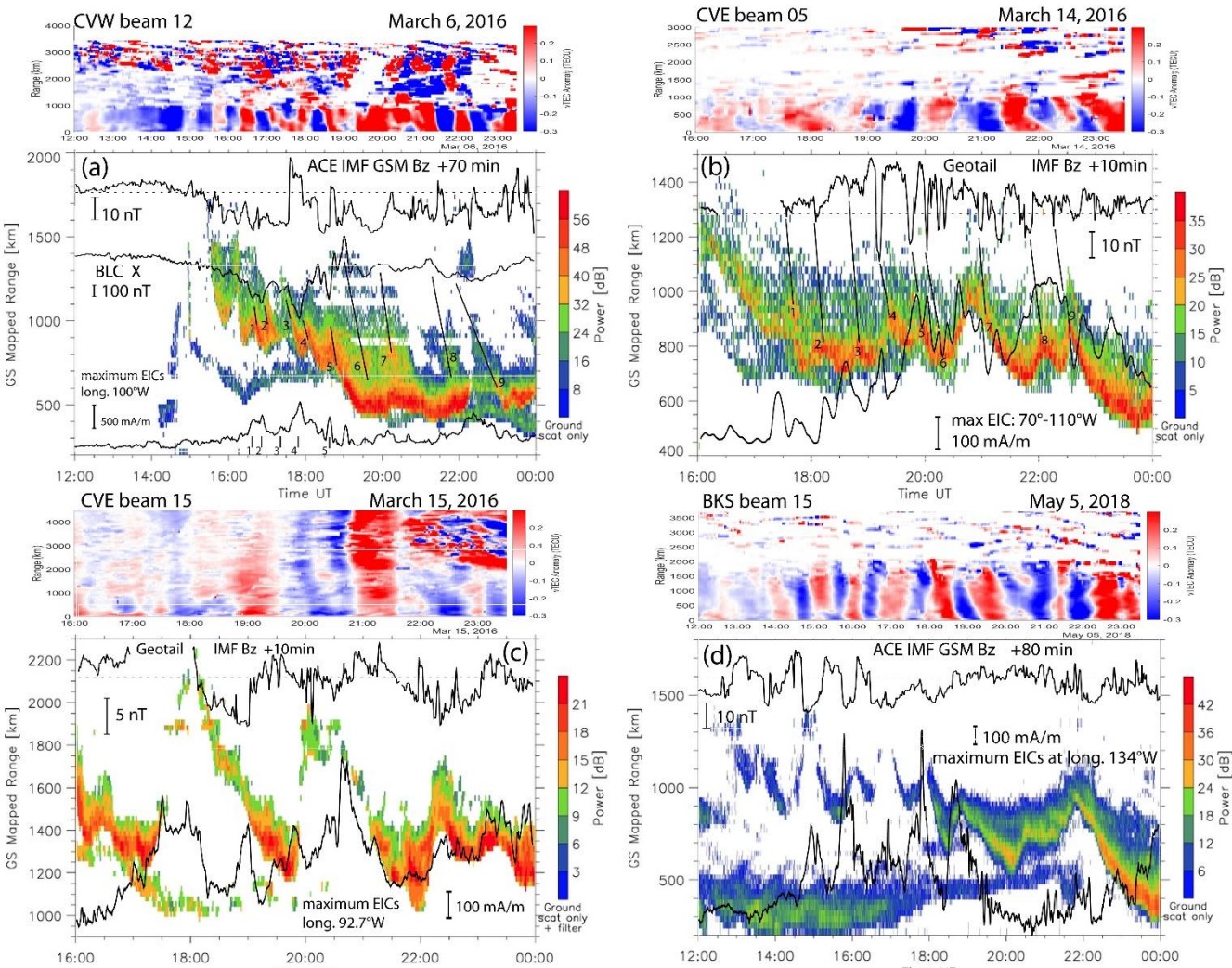

**Figure 9:** TIDs observed in the radar ground-scatter power and the detrended TEC along **(a)** the CVW radar beam 12 on March 6, 2016, **(b)** the CVE radar beam 5 looking on March 14, 2016, **(c)** the CVE radar beam 15 on March 15, 2016, and **(d)** the BKS radar beam 15 on May 5, 2018. The time-shifted IMF $B_z$, the time series of the latitudinal maxima in EICs at given longitudes, and in panel **(a)**, the X component of the ground magnetic field measured in Baker Lake (BLC), are superposed.

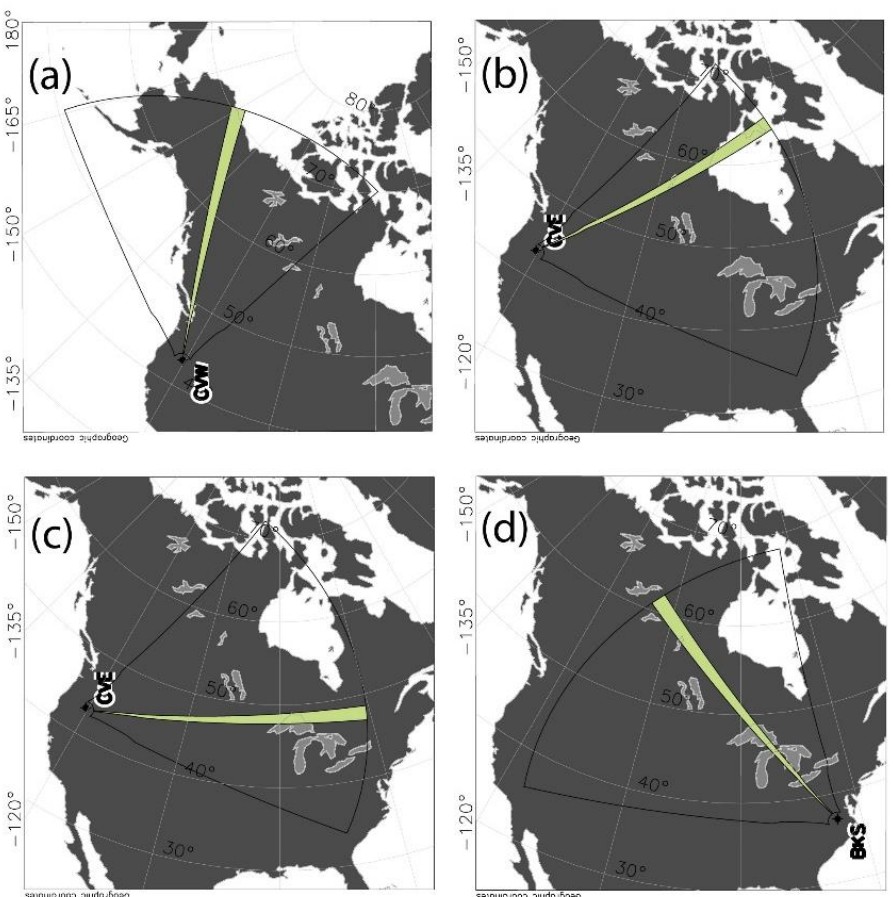

**Figure 10:** Radar FoVs and selected beams of the **(a)** Christmas Vallee West (CVW) radar beam 12, **(b)** Christmas Vallee East (CVE) radar beam 5 **(c)** the CVE radar beam 15, and **(d)** BKS radar beam 15 used in Fig. 9.

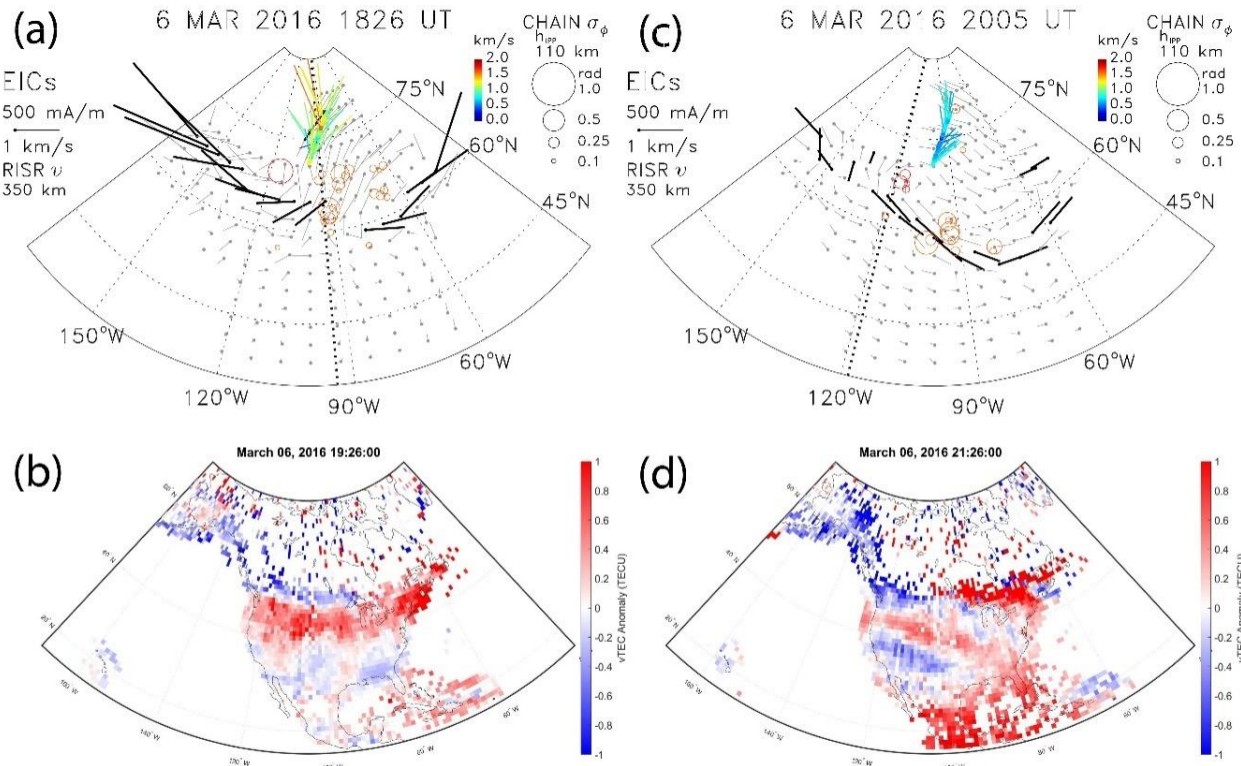

**Figure 11: (a, c)** The horizontal EICs with the latitudinal maxima highlighted in bold and the CHAIN GPS IPPs with enhanced phase variation values, $\sigma_\Phi$, (scaled circles). **(b, d)** The detrended vTEC maps at later times on March 6, 2016.

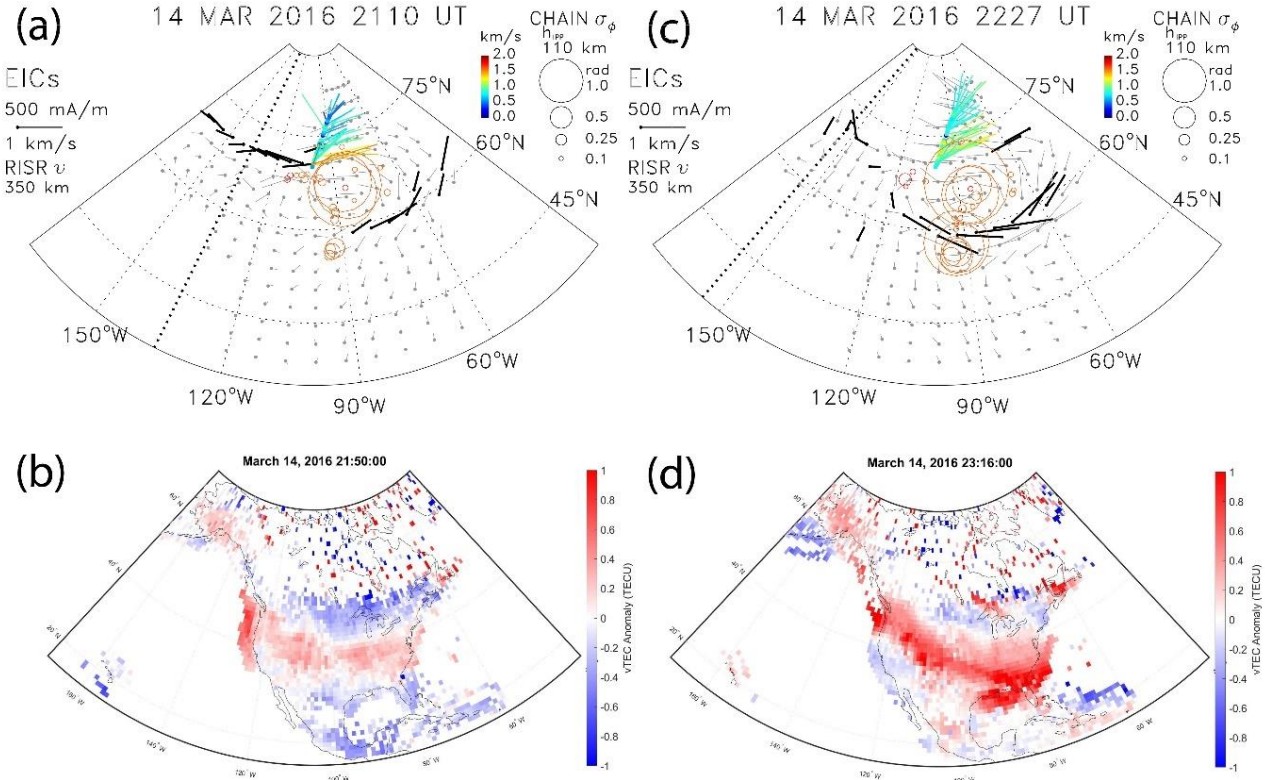

**Figure 12: (a, c)** The horizontal EICs with the latitudinal maxima highlighted in bold and the CHAIN GPS IPPs with enhanced phase variation values, $\sigma_\Phi$, (scaled circles). **(b, d)** The detrended vTEC maps at later times on March 14, 2016.

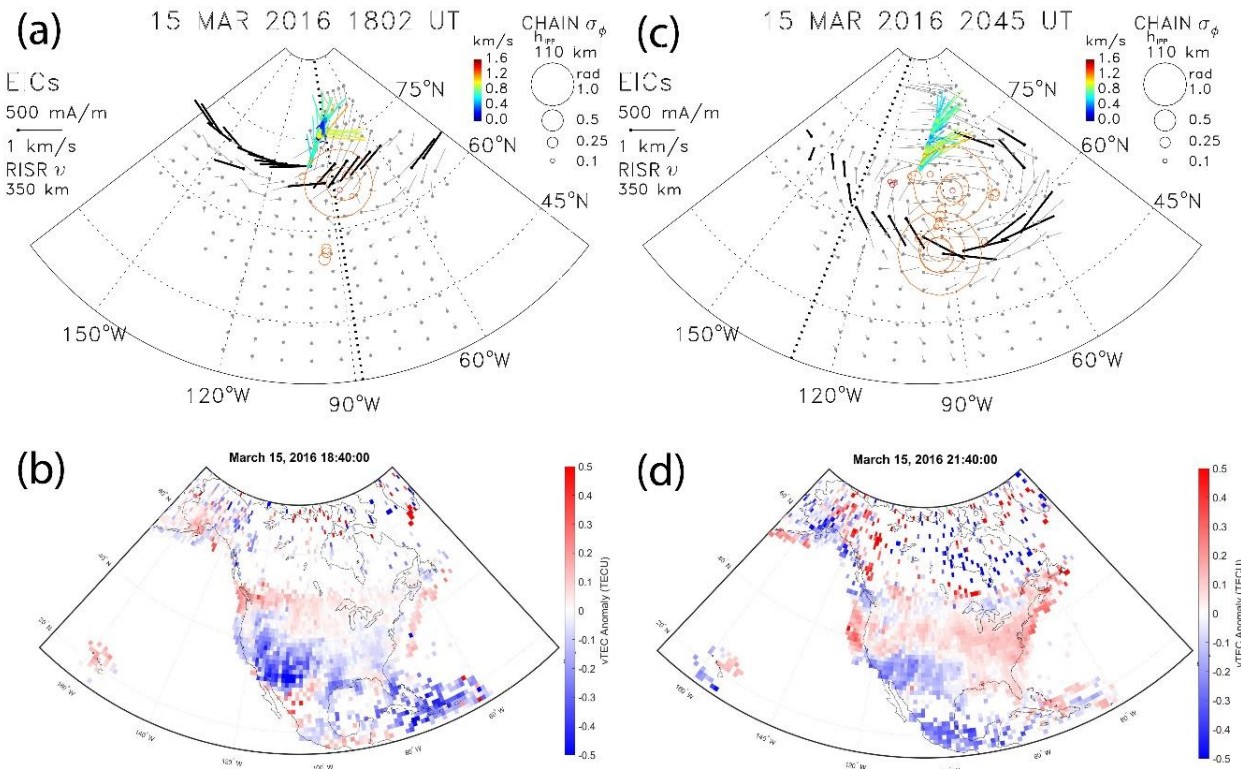

**Figure 13: (a-b)** The horizontal EICs with the latitudinal maxima at each longitude grid highlighted in bold and the CHAIN GPS IPPs with enhanced phase variation values, $\sigma_\Phi$, (scaled circles). **(c-d)** The detrended vTEC maps showing LSTIDs at later times on March 15, 2016.

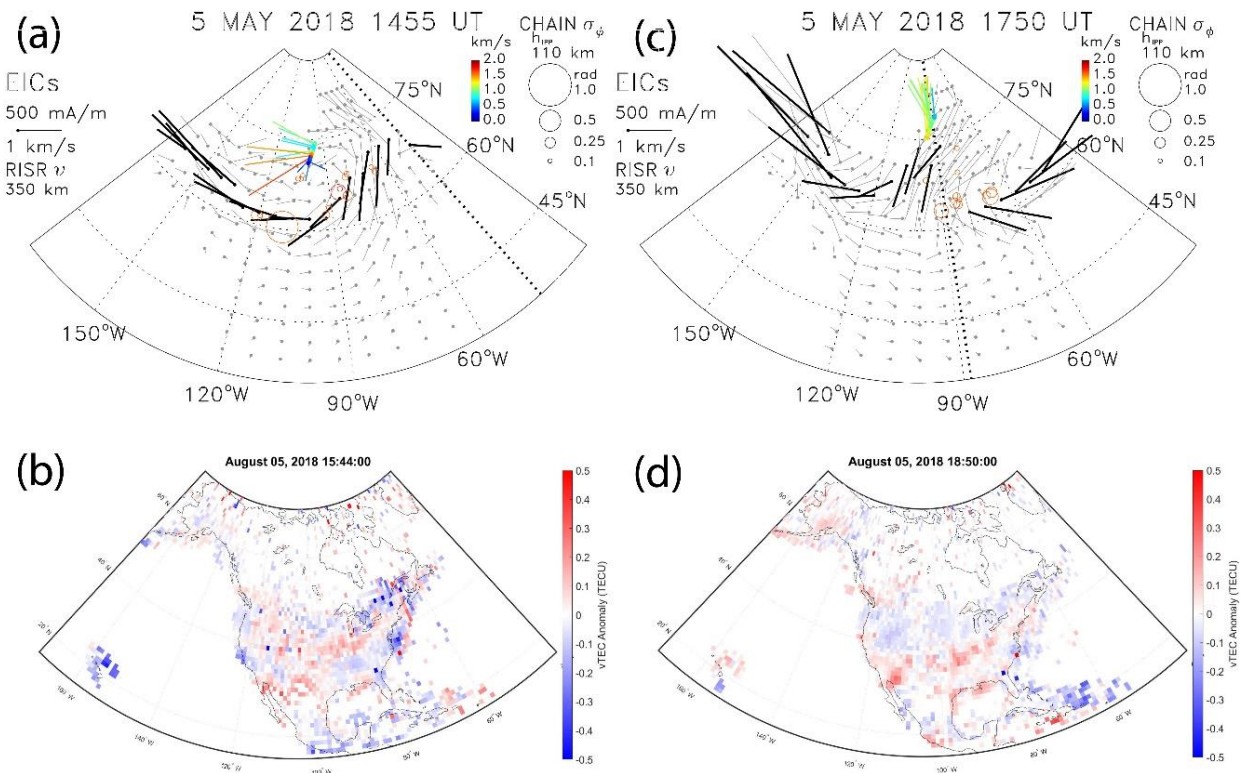

**Figure 14: (a-b)** The horizontal EICs with the latitudinal maxima at each longitude grid highlighted in bold and the CHAIN GPS IPPs with enhanced phase variation values, $\sigma_\Phi$, (scaled circles). **(c-d)** The detrended vTEC maps showing LSTIDs at later times on May 5, 2018.

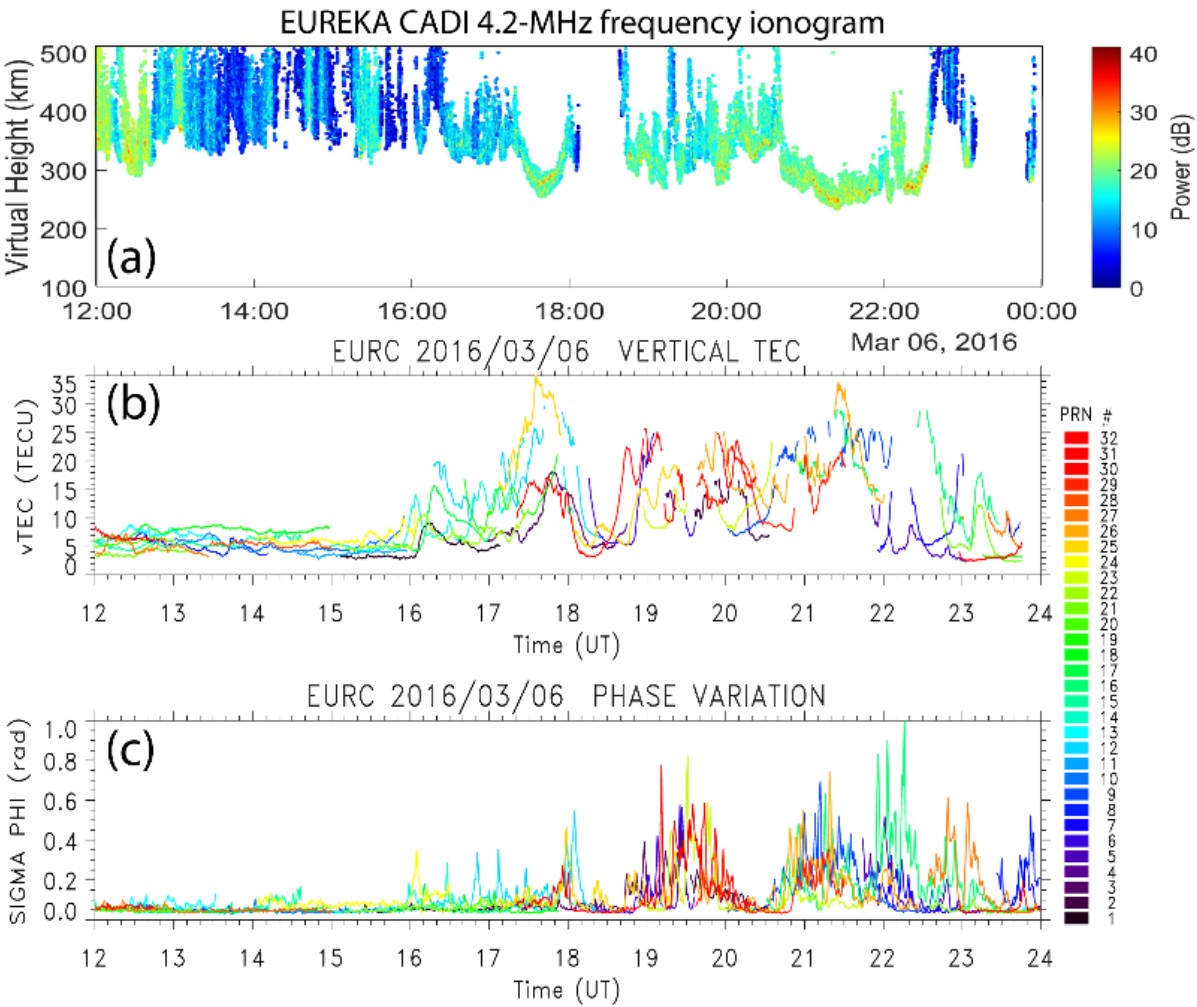

**Figure 15: (a)** The fixed 4.2-MHz frequency ionogram, the GPS **(b)** vTec, and **(c)** phase variation observed from Eureka on March 6, 2016.

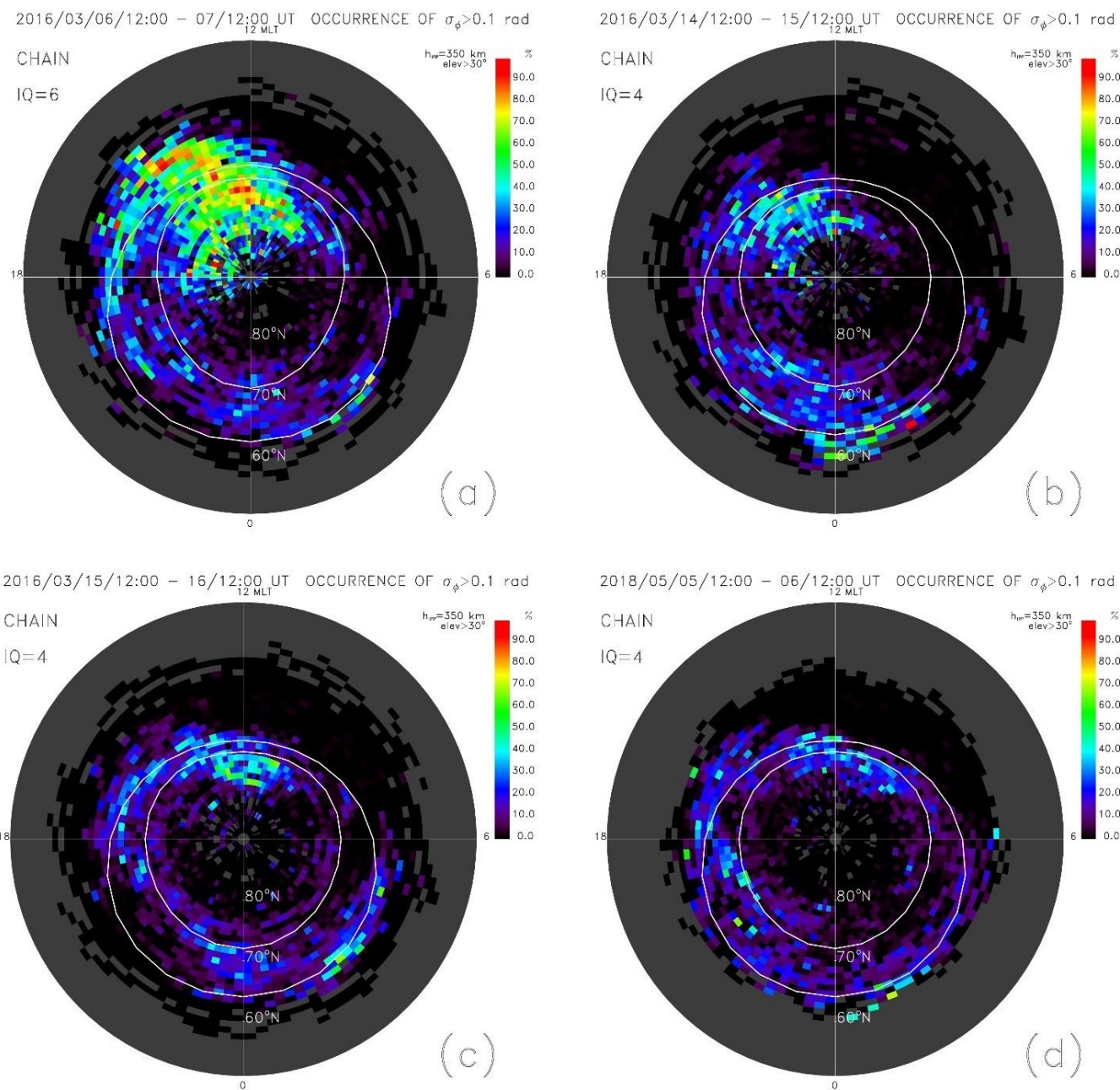

**Figure 16:** The percentage occurrence of the GPS phase variation $\sigma_\Phi > 0.1$ rad mapped in coordinates of AACGM latitude
and MLT during geomagnetic storms on **(a)** March 6, **(b)** March 14, **(c)** March 15, 2016, and **(d)** May 5, 2018. Boundaries of
the statistical auroral oval are shown.

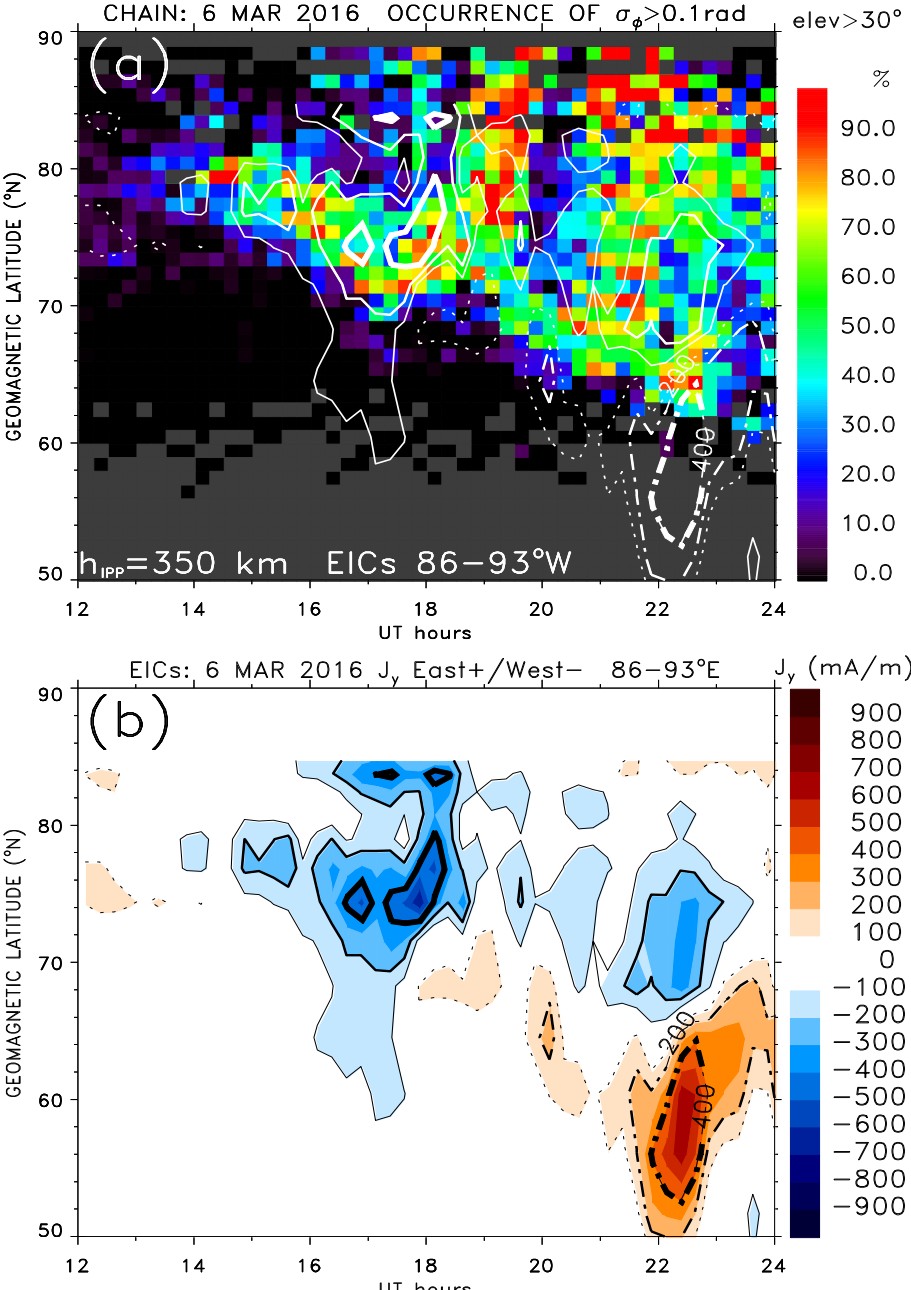

**Figure 17: (a)** The GPS phase variation occurrence as a function of AAGCM latitude and UT on March 6, 2016. Contours of the westward and eastward EICs are shown in white solid and broken lines, respectively. **(b)** Westward and eastward EICs are shown in blue and brown shades.

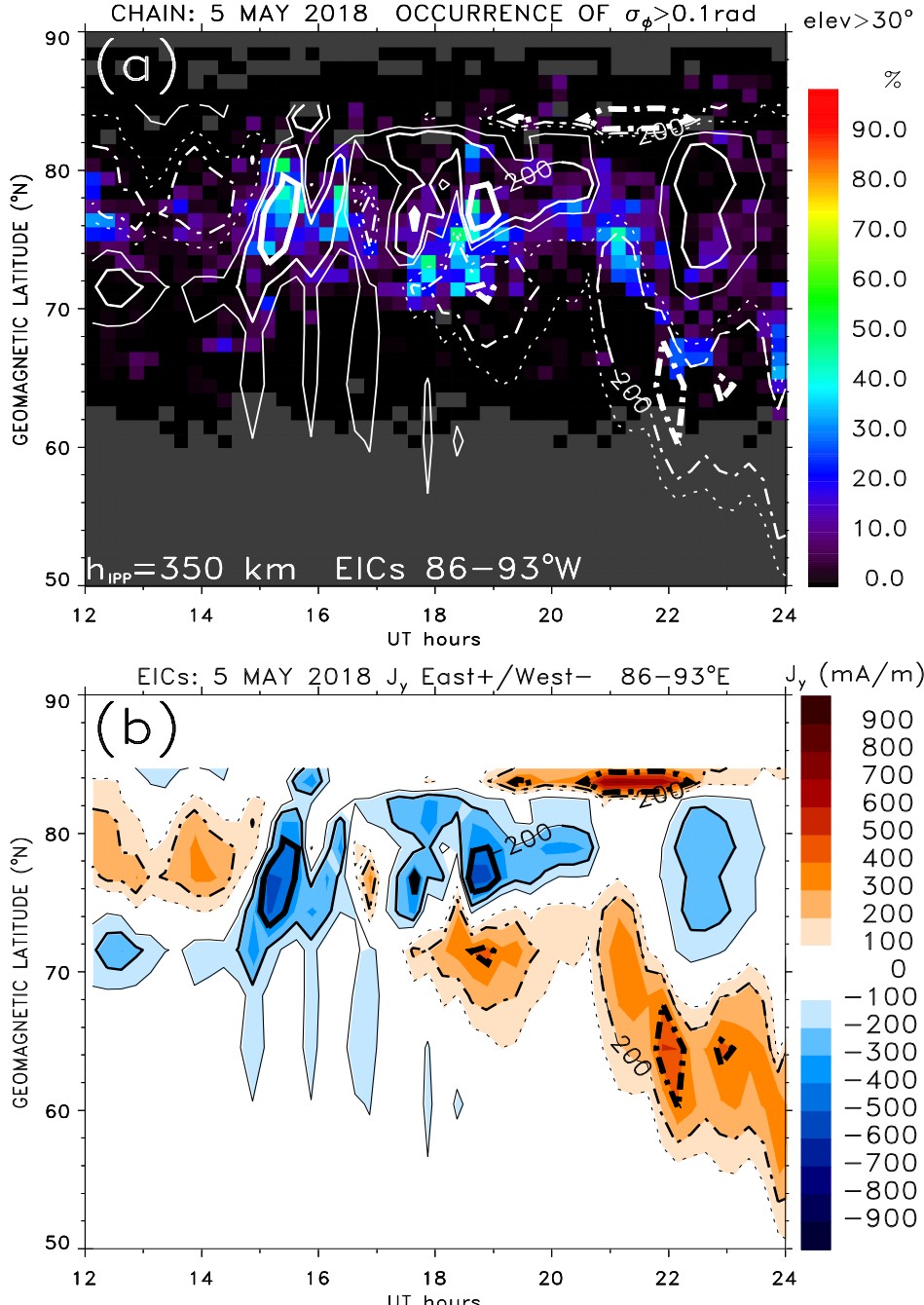

915

**Figure 18:** The same as Fig. 17 but for May 5, 2018.