# Peer review of "Multi-instrument observations of polar cap patches and traveling ionospheric disturbances generated by solar wind Alfvén waves coupling to the dayside magnetosphere"

_Annales Geophysicae, 2022_

## Author Comment (AC1)

**RC1:** 'Comment on angeo-2022-14', Anonymous Referee #1, 29 Jun 2022

**Replies to Reviewer 1**

Reviewer's report on the manuscript by Prykryl et al. titled "Multi-instrument observations of polar cap patches and traveling ionospheric disturbances generated by solar wind Alfvén waves coupling to the dayside magnetosphere" (manuscript #: angeo-2022-14)

In this manuscript, the authors try to show a close relationship between the solar wind Alfvén waves and the polar cap patches / traveling ionospheric disturbances, using multiple events based on the observation by the RISR IS radar, ground-based magnetometers, GNSS receivers, and SuperDARN radars. The topic itself is scientifically interesting, although several points should be improved. I consider that the following points should be addressed and revised before the manuscript is ready for publication in the Annales Geophysicae journal.

*Reply: We appreciate your comments and suggestions for improvements. We considered them all and revised the manuscript accordingly. Below we provide our replies (in italics) to individual points.*

Overall comments:

1. Interpretation of the data. The authors demonstrate that the TIDs are generated only by the Joule Heating due to the dayside ionospheric currents. These days it is not a generally accepted idea. The TIDs are highly correlated with the lower atmospheric disturbances (e.g., Frissell et al., 2016). I am also surprised to see that the manuscript includes minimal discussion. The authors should add the discussion section, cite the related papers such as those mentioned in this report, and discuss the differences between the current events and the previous studies.

Recent progress of the TID studies using the SuperDARN is summarized in the review paper by Nishitani et al. (2019). I recommend that the authors check this paper.

*Reply 1: Although some discussion was distributed in specific sections, we fully agree that the Discussion section is needed to summarize and compare our results with previous studies. It is now included in Section 4. This includes references to papers that provided various interpretations of TIDs and pointed to sources other than high-latitude ionospheric currents. Of course, the above papers, particularly the comprehensive 2019 paper, should not have been missed. It is now quoted several times throughout the manuscript. However, we believe that the presented case studies of equatorward propagating TIDs observed by SuperDARN and GNSS receivers clearly point to sources of gravity waves in the dayside high-latitude ionosphere, the ionospheric currents modulated by solar wind Alfvén waves. This is particularly evident in Fig. 9d and is consistent with previously published results (Prikryl et al., 2005).*

2. Definition of the LSTID / MSTID. The manuscript states that the LSTIDs have greater than 10000 km. I am not certain how the authors can distinguish between LSTIDs and MSTIDs in Figure 9. The authors need to add a more detailed description. In addition, I am not sure whether the LSTIDs observed by the SuperDARN radar and pointed out in the text are actually LSTIDs. Lines 261-262 state, "Figs. 9a-d show TIDs observed in the detrended vertical vTEC and the radar ground-scatter power focused and defocused by

TIDs moving equatorward." The LSTID wavelength greater than 1000 km cannot produce focusing / defocusing of the radar waves. LSTIDs are supposed to be observed in the Doppler velocities of the ground scatter data. For example, Hayashi et al. (2010) showed that the SuperDARN Doppler velocities changes in the ground scatter data are consistent with the GNSS TEC data in the framework of the propagation of atmospheric gravity waves.

*Reply 2: In the Introduction we referred to previous studies that provided definition of LSTIDs/MSTIDs: "Large-scale TIDs (LSTIDs) generally propagate at speeds between 400 and 1,000 ms−1, have wavelengths greater than 1000 km, and periods of 30 - 180 min, while medium-scale TIDs (MSTIDs) tend to propagate at speeds of 250 - 1,000 ms−1, and have wavelengths of several hundred kilometers and periods of 15 - 60 min (Francis, 1975; Hunsucker, 1982; Zhang et al., 2019)."*

*We agree that it may not be possible to strictly distinguish between MSTIDs and LSTIDs, because of a continuum of sizes and periods. But we disagree with the statement that LSTIDs with wavelengths greater than 1000 km cannot focus HF radio waves. The electron enhancements, particularly when slanted as in TIDs, would certainly refract the radio waves, and focus them to produce enhanced ground scatter power. Of course, we agree that TIDs can also be observed in the variations of Doppler velocities changes in the ground scatter data as shown and correlated with the GNSS TEC in the quoted paper by Hayashi et al. (2010, their Figs. 4 and 5). However, their equatorward propagating LSTIDs (Events 1 and 2) can also be clearly identified in the ground scatter power ([https://cicr.isee.nagoya-u.ac.jp/web1/superdarn/sddata/hokql/gif/hok/2006/bm00/20061215_hok_bm00_ql.gif](https://cicr.isee.nagoya-u.ac.jp/web1/superdarn/sddata/hokql/gif/hok/2006/bm00/20061215_hok_bm00_ql.gif)), although not shown or discussed by the authors. This is in contrast with their poleward propagating LSTID Event 3 that they observed both in TEC and Doppler velocity, but that does not seem to be observed in the ground scatter power.*

3. Lines 144-145 say, "Fig. 3 shows the ionospheric currents (EICs) mapped in geographic coordinates…." It is strange that later in the manuscript, Figures 16-18 are plotted in AACGM (geomagnetic coordinates). I do not understand why the authors plot the same (e.g., EIC) data in different coordinates in one manuscript. It will cause serious confusion among the readers. I strongly recommend plotting the data in the same coordinate system.

*Reply 3: Figures 16-18 that can be compared with Fig. 4 are shown in geomagnetic coordinates, which are also used in our previous study (Prikryl et al., 2016) that we are citing. Other figures in the present paper, particularly the RISR data (Fig. 2), PIFs (Figs. 6 and 7), EICs (Figs. 3, 5, etc.) and TEC maps (Figs. 11 to 14) are all using geographic coordinates. Unfortunately, we cannot provide all figures in AACGM coordinates, and we do not believe it is necessary to do that.*

Individual comments:

Lines 128-129 and Figure 2: Please describe the RISR-C and RISR-N field of views (beam positions). Otherwise, the readers cannot understand what the authors mean.

*Reply: Reference to Gillies et al., (2016; see, their Fig. 1) is provided. Also, geographic maps (Fig. 3, etc) show RISR velocity vectors.*

Lines 138-140: "The first few patches (enhancements in Ne) that were observed by RISR-N between 16:00 and 17:00 UT were not detected by RISR-C (Fig. 2a). This implies that the cusp was in the RISR-C FoV since polar patches are known to be produced by flow channels in the cusp." I do not understand these sentences. Maybe something is wrong. Please check.

**Reply**: *The first sentence is modified: "The first few patches (enhancements in $N_e$) started to be observed by RISR-N north of 75°N between 16:00 and 17:00 UT and were not detected by RISR-C (Fig. 2a)."*

Lines 146-147: "The GPS ionospheric pierce points (IPPs) at 110 km shown as circles scaled by the CHAIN GPS phase variation values, σΦ, are discussed in Section 3.3.3." – I wonder why the authors set the pierce points at 110 km. Obviously, the electron density is higher in the F-region than in the E-region, and so is the amplitude of scintillations. By the way, I cannot find section 3.3.3.

**Reply**: *Both the EICs and IPPs are mapped at 110-km altitude to show that, in the auroral zone, IPPs of strong GPS phase scintillation are largely collocated with the electrojet currents (Prikryl et al., 2016; 2021). This is because, as discussed by the latter authors, there is a relation between vertical currents (Jz; not shown in the present paper) and strong GPS phase scintillation (variations) that map to upward or downward Jz, or near the reversal boundaries between downward and upward Jz, as would be expected for scintillation being caused by ionization due to precipitating electrons, which can maximize at lower than F-region altitudes. In section 3.3, the focus is on polar cap patches, so the occurrence maps of GPS variation occurrence are shown for 350-km altitude. The question of the actual altitude where the GPS scintillation originates has not been fully resolved.*

Line 185: "were" – is it "which were"?

**Reply**: *Agree. This is now corrected.*

Figure 9 and Lines 266-267 (as well as other corresponding lines): Are the ground scatter ranges plotted the same way as the ionospheric scatter? If so, it will cause a severe misunderstanding among the readers. If the ground scatter comes from a 1-hop propagation mode, then the focusing / defocusing point should be the mid-point between the radar and the backscatter region (for 2+ hops the geometry becomes more complicated). It is not appropriate to plot the SuperDARN echo data with the range set to the backscatter point, together with the GNSS TEC data with the same range.

**Reply**: *Agree. Fig. 9 is modified showing the ground scatter mapped range using mapping discussed by Bristow et al. (1994) and Frissell et al. (2014).*

Figure 4 caption: There is no description of the SuperDARN convection map.

**Reply**: *The missed description is now added.*

References:

Prikryl, P., et al. (2016), GPS phase scintillation at high latitudes during the geomagnetic storm of 17–18 March 2015, J. Geophys. Res. Space Physics, 121, doi:10.1002/2016JA023171.

Prikryl, P., J. M. Weygand, R. Ghoddousi-Fard, P. T. Jayachandran, D. R Themens, A. M. McCaffrey, B. S. R. Kunduri, L. Nikitina, Temporal and spatial variations of GPS TEC and phase during auroral substorms and breakups, Polar Science, Vol. 28, 2021, 100602. https://doi.org/10.1016/j.polar.2020.100602

Frissell, N. A., Baker, J. B. H., Ruohoniemi, J. M., Gerrard, A. J., Miller, E. S., Marini, J. P., West, M. L., and Bristow, W. A. (2014), Climatology of medium-scale traveling ionospheric disturbances observed by the midlatitude Blackstone SuperDARN radar, *J. Geophys. Res. Space Physics*, 119, 7679– 7697, doi:10.1002/2014JA019870.

Frissell NA, Baker JBH, Ruohoniemi JM, Greenwald RA, Gerrard AJ, Miller ES, West ML (2016) Sources and characteristics of medium-scale traveling ionospheric disturbances observed by high-frequency radars in the North American sector. J Geophys Res Space Physics 121:3722–3739. https://doi.org/10.1002/2015JA022168

Hayashi H, Nishitani N, Ogawa T, Otsuka Y, Tsugawa T, Hosokawa K, Saito A (2010) Large-scale traveling ionospheric disturbance observed by SuperDARN Hokkaido HF radar and GPS networks on 15 December 2006. J Geophys Res 115:A06309. https://doi.org/10.1029/2009JA014297

Nishitani, N., Ruohoniemi, J.M., Lester, M. et al. Review of the accomplishments of mid-latitude Super Dual Auroral Radar Network (SuperDARN) HF radars. Prog Earth Planet Sci 6, 27 (2019). https://doi.org/10.1186/s40645-019-0270-5

---

## Author Comment (AC2)

**Replies to Reviewer 2**

Review of "Multi-instrument observations of polar cap patches and traveling ionospheric disturbances generated by solar wind Alfven waves coupling to the dayside magnetosphere" by Prikryl, Gillies, Themens, Weygard, Thomas and Chakraborty

The paper is well written, contains interesting new results and should be published in Annales Geophysicae after suitable revision.

*Reply: We appreciate your comments and suggestions. They helped us to clarify the points are trying to make, to improve referencing relevant papers, and hopefully to improve the manuscript in general. Below we provide our replies (in italics) to individual points. The manuscript is revised accordingly.*

**Main Comments**

There are two distinct parts to solar wind high speed streams. At the leading edge where the high speed stream interacts with the upstream slow speed stream, a "corotating interaction region" or CIR (GRL 3, 3, 137-140, 1976; JGR 100, A11, 21717-21733, 1995) forms. CIRs have both high magnetic fields and high plasma densities, higher than the following high speed stream proper. The Alfven wave amplitudes are also higher inside the CIR due to the compression (GRL, 22, 23, 3397-3400, 1995). It will be interesting for the AG readership to know where your effects are strongest, associated with the CIR or the high speed stream proper. Also since the plasma densities inside the CIR are high, can this play a role in magnetic reconnection and the tongues of ionization?

*Reply: Yes, we agree that the effect should be stronger for CIRs, which we focus on in the manuscript. We now discuss this in the Introduction and introductory paragraph of Section 3 and provide relevant references.*

Introduction Section. I doubt that Jim Dungey (1961) intended to imply that the interplanetary magnetic field remained southward and there occurred a steady state of energy input into the magnetosphere. This statement should be reworded a bit to remove this implication.

*Reply: The reference to Dungey (1961) is replaced by appropriate references (Russell and Elphic, 1978, 1979).*

Short duration (~30 min to 1 hr) southward magnetic fields causes substorms (PSS, 12, 273-282, 1964; JGR, 77, 16, 2970, 1972; JGR, 78, 4, 617-629, 1973). Longer duration (hrs) southward fields cause magnetic storms (JGR, 99, A4, 5771-5792, 1994; JGR 113, A05221, doi:10.1029/2007JA012744, 2008). Southward component interplanetary fields associated with Alfven waves in either CIRs or high speed streams have been shown to do the same, cause substorms and DP2 events (JGR, 73, 11, 5549-5559, 1958; JGR, 95, A3, 2241-2252, 1990; JGR 100, A11, 21717-21733, 1995; JASTP 66, 167-176, 2004). In the JGR 2000 paper it was noted that southward IMFs with durations less than 15 min were not geoeffective. Can you please mention (roughly) the duration of the southward components of the Alfven waves?

*Reply:* *Your references mostly point to causes of substorms and magnetic storms. Although the events we describe in the manuscript occurred during the growth phase of geomagnetic storms, we focus on the solar wind coupling on the dayside. The durations of the southward component of the Alfven waves, as can be seen in Fig. 9, varied from a few minutes to a few hours. We believe that the southward turnings, as well as the IMF By duskward deflections, play a role in the onset of flows in the cusp. Of course, the anti-sunward flows will intensify, and last longer, when Bz remain southward longer. This will certainly influence the geo-effectiveness, and on the substorm and magnetic storm developments, as more and more newly opened magnetic flux is carried over to the nightside. But we are focussing on the immediate response in the cusp and ionospheric signatures of FTEs. We have added these sentences in the Introduction:*

*"Solar wind Alfvén waves (Belcher and Davis, 1971) that couple to the magnetosphere-ionosphere system are associated with high-intensity long-duration continuous auroral activity (HILDCAA) (Tsurutani and Gonzalez, 1987; Tsurutani et al., 1990). Spacecraft observations of the polar cap and auroral zone noted auroral patches during HILDCAA intervals due to the southward component of Alfvén waves causing reconnection (Guernieri et al., 2004; Guernieri 2006). The durations of the southward component of Alfvén waves have an impact on the geo-effectiveness, and on the substorm and magnetic storm developments. However, in this paper we focus on the immediate dayside ionospheric response to the IMF during the impact of corotating interaction regions at the leading edge of high-speed streams."*

The ionospheric currents that you mention most certainly must be DP2 currents. Please quote and discuss.

*Reply:* *Thank you for this important comment. In the new Section 4 Discussion we quote relevant papers and briefly discuss the currents, including DP2.*

Line 52. It should be noted that spacecraft observations of the polar cap and auroral zone noted auroral patches during HILDCAA intervals ( southward component of Alfven waves in solar wind high speed streams causing reconnection). Please see p235-243 in AGU mon. 167, 2006; Substorms 7: Proceedings of the 7th International Conference on Substorms,edited by N. Ganushkina and T.I. Pulkkinen, 1, 67, 2004. These papers should be quoted.

*Reply:* *The sentence is modified, and the references are added.*

Line 113. The reference to the HCS discovery should be quoted here. It is JGR 83, 717, 1978.

*Reply:* *The reference is included. Thank you.*

---

## Author Comment (AC3)

**RC4**: 'Reply on AC1', Anonymous Referee #1, 13 Jul 2022

**Response to Reviewer 1**

Since I have not seen a revised manuscript, I just comment on one of the authors' replies.

***Reply 1:*** *Unfortunately, we have been advised not to upload the revised manuscript with our replies to comments. Here we include at least one part of the revision that is relevant to the Reviewer comments:*

***"4 Discussion***

*The presented multi-instrument observations of polar cap patches in the Canadian Arctic are consistent with previously published results (e.g., Provan et al, 1998) that support the accepted model of polar patch formation (Cowley and Lockwood, 1992). Transient azimuthal flows in the cusp that resulted in the formation of polar cap patches were associated with the IMF $B_y$ fluctuations due to solar wind Alfvén waves. Pulsed ionospheric flows modulated by solar wind Alfvén waves followed by polar cap patches were previously observed (Prikryl et al., 1999; 2002).*

*The large-amplitude solar wind Alfvén waves in the CIRs at the leading edge of HSSs also modulated the ionospheric currents that were estimated from the ground-based magnetometer data using an inversion technique. The ionospheric currents have been recognized as sources of AGWs causing TIDs. Of course, AGWs/TIDs can be generated by various other sources, including tropospheric weather systems (Bertin et al., 1975, 1978; Waldock and Jones, 1987; Oliver et al., 1997; Nishioka et al. 2013), polar vortex (Frissell et al., 2016), volcanic eruptions, earthquakes, and tsunamis (e.g., Nishitani et al., 2019; Themens et al., 2022), as well as phenomena associated with ion-neutral interactions (Nishitani et al., 2019). However, the case studies of equatorward propagating TIDs observed by SuperDARN and GNSS receivers presented in this paper clearly point to dayside ionospheric currents modulated by solar wind Alfvén waves. This is consistent with the previously published results (Prikryl et al., 2005).*

*Milan et al. (2017; see, their Fig. 2) reviewed the morphology and dynamics of the electrical current systems of the terrestrial magnetosphere and ionosphere that include DP1, DP2 and DPY currents. The patch formation has been associated with the By-modulated DPY currents (Hall currents associated with FCEs) (Friis-Christensen and Wilhjelm, 1975; Clauer et al., 1995; Stauning et al. 1994, 1995; Prikryl et al. 1999). In the high conductance auroral zone, Hall currents form the eastward and westward auroral electrojets, and the corresponding magnetic perturbations on the ground associated with these Hall currents, are known as the DP1 and DP2 patterns. However, this paper is concerned with the dayside currents, so the TIDs were caused primarily by the DP2 current intensifications."*

>We agree that it may not be possible to strictly distinguish between MSTIDs and LSTIDs, because of a continuum of sizes and periods. But we disagree with the statement that LSTIDs with wavelengths greater than 1000 km cannot focus HF radio waves. The electron enhancements, particularly when slanted as in TIDs, would certainly refract the radio waves, and focus them to produce enhanced ground scatter power. Of course, we agree that TIDs can also be observed in the variations of Doppler velocities changes in the ground scatter data as shown and correlated with the GNSS TEC in the quoted paper by Hayashi et al. (2010, their Figs. 4 and 5). However, their equatorward propagating LSTIDs (Events 1 and

2) can also be clearly identified in the ground scatter power (https://cicr.isee.nagoya-u.ac.jp/web1/superdarn/sddata/hokql/gif/hok/2006/bm00/20061215_hok_bm00_ql.gif), although not shown or discussed by the authors. This is in contrast with their poleward propagating LSTID Event 3 that they observed both in TEC and Doppler velocity, but that does not seem to be observed in the ground scatter power.

I agree that LSTIDs can divert the radar waves forth and back owing to the tilted isopycnic surface. However, it does not mean the radar waves are focused or defocused. I can see that the plot the authors showed indicates maximum echo power region moves away and toward the radar in association with the LSTID. Still, it cannot be called focusing / defocusing of the radar wave packets at all because it does not show the propagating structure (just forth and back). It is thus appropriate to use words other than focusing / defocusing, such as diverting the maximum echo power region forth and back. If the authors disagree, they need to show that the focusing / defocusing can modulate the echo power significantly with the gravity waves with a wavelength of more than 1000 km using the HF ray path tracing technique.

**Reply 2:** *We agree, and we have now clarified this in the revised manuscript:*

*"In the case of LSTIDs with a wavelength of more than 1000 km the tilted isopycnic surfaces divert the refracted radio waves back and forth thus modulating the range of the ground scatter."*

---

## Author Response (AR2)

**Reviewer 1**

Lines 30-33. The authors misinterpreted my previous comment about "a steady state phenomenon" and the reference to Dungey, 1961. I think when one discusses magnetic reconnection, one should have the Dungey reference there. His picture/theory was the first for the magnetosphere and it should definitely be cited. I was just reacting to the comment about "steady state convection". One need not use that phrase. Your current phrase "thought to be either continuous (quasi steady) or pulsed (impulse)" is also not needed either.

I suggest putting back the Dungey, 1961 reference and adding Tsurutani and Meng, 1972 to the list. The latter will take care of observations to substorms and the AE indices.

*Line 29-31 now revised*: *"The magnetic reconnection on the dayside magnetopause leads to open magnetic flux carried over the polar cap to the magnetotail (Dungey, 1961; Tsurutani and Meng, 1972; Russell and Elphic, 1978, 1979; Provan et al., 1998)."*

I had mentioned (with references) that short duration (less than ~15 min) southward magnetic fields had not shown any geomagnetic effects. The flux transfer events are probably seconds in duration, so it is unclear whether they could cause your polar patches or not. One has no idea how long they would have to last to make a polar patch. But maybe leave those references?

*We agree that some of the FTE signatures observed by RISR were short, but the FoV of RISR is limited.*

**Reviewer 2**

Lines 142-143: Please describe the maximum / minimum latitudes of the RISR-C and RISR-N fields of views. The authors are not kind to the audience because they require us to check another paper's figure, and even I look at Figure 1 of Gillies et al. (2016), it is not easy to connect their Figure 1 and the current Figure 2. Please clarify which parts correspond to RISR-C or RISR-N, which are mentioned several times in the current manuscript.

We have clarified the geographical areas covered by RISR-N and RISR-C:

Lines 76-77: "*The Resolute Bay Incoherent Scatter Radars (RISR) covering latitudes from 75° to 81°N (RISR-N) and from 69° to 75°N (RISR-C) are located at a geographic latitude of 74.70°N and geographic longitude of 94.83°W.*"

*Lines 145-146*: *"Fig. 2a shows Ne and anti-sunward Ve averaged over the longitude span of the RISR-N beams (from 75° to 100°W) and RISR-C beams (from 93° to 107°W...)"* We retained the reference to Gillies et al. (2016; see, their Fig. 1).

Line 182 and Figure 4 caption should describe that the convection map based on the SuperDARN observations.

Line 182: "global ionospheric convection map: should be "SuperDARN global convection map"

Figure 4 caption: "Ionospheric convection and potential maps" should be "SuperDARN ionospheric convection/potential map"

*Line 182 and Figure 4 caption* *now clarify that the convection/potential maps are the SuperDARN observations.*